# VIEWSPATIAL-BENCH: EVALUATING MULTI-PERSPECTIVE SPATIAL LOCALIZATION IN VISION-LANGUAGE MODELS

## ABSTRACT

Vision-language models (VLMs) have demonstrated remarkable capabilities in understanding and reasoning about visual content, but significant challenges persist in tasks requiring cross-viewpoint understanding and spatial reasoning. We identify a critical limitation: current VLMs excel primarily at egocentric spatial reasoning (from the camera's perspective) but fail to generalize to allocentric viewpoints when required to adopt another entity's spatial frame of reference. We introduce ViewSpatial-Bench, the most comprehensive benchmark designed specifically for multi-viewpoint spatial localization recognition evaluation across five distinct task types, supported by an automated 3D annotation pipeline that generates precise directional labels. Comprehensive evaluation of diverse VLMs on ViewSpatial-Bench reveals a significant performance disparity: models demonstrate reasonable performance on camera-perspective tasks but exhibit reduced accuracy when reasoning from a human viewpoint. By fine-tuning VLMs on our multi-perspective spatial dataset, we achieve an overall performance improvement of 46.24% across tasks, highlighting the efficacy of our approach. Our work establishes a crucial benchmark for spatial intelligence in embodied AI systems and provides empirical evidence that modeling 3D spatial relationships enhances VLMs' corresponding spatial comprehension capabilities.

## 1 INTRODUCTION

While Vision-Language Models (VLMs) demonstrate remarkable capabilities in visual content understanding and reasoning (Chen et al., 2024a; Cheng et al., 2024; Song et al., 2024), they exhibit significant limitations when confronted with complex tasks requiring cross-viewpoint comprehension and spatial reasoning (Shiri et al., 2024; Stogiannidis et al., 2025). Specifically, current VLMs perform adequately in egocentric spatial judgments but struggle to interpret and reason about spatial relationships from alternative entity perspectives (Lee et al., 2025). This constraint substantially impedes the performance of the model in practical application scenarios.

Humans naturally understand spatial relationships from multiple perspectives. When interacting with others, we effortlessly adopt their viewpoints to interpret spatial references: intuitively distinguishing between "*the cup on my left*" and "*the cup on your left*" without conscious effort. This perspective-taking ability enables seamless communication in physical spaces and forms the foundation for successful collaborative interactions. In contrast, current VLMs operate primarily within an egocentric reference frame, where spatial reasoning is entirely anchored to the camera's perspective (Paz-Argaman et al., 2024).

This issue is particularly prominent in embodied interaction scenarios. When a person asks a robot "Can you pass the mug on my right?", they expect the robot to identify the target object from their perspective rather than the robot's own. This ability to reason spatially from different viewpoints, known in cognitive science as "perspective-taking," represents a critical capability for human-machine interaction, spatial navigation (Zhao et al., 2024), and multi-agents collaboration (Feng et al., 2025). Crucially, this challenge becomes significantly more complex in three-dimensional environments,

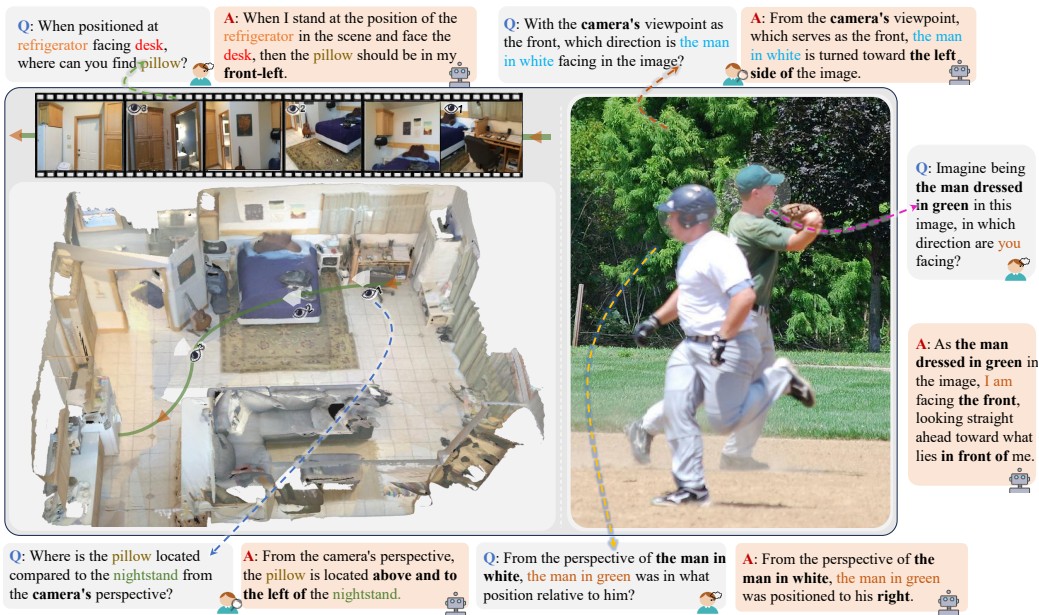

Figure 1: ViewSpatial-Bench for multi-perspective spatial reasoning. Our benchmark evaluates spatial localization capabilities from both camera and human perspectives across five task types.

where viewpoint transformation involves not only changes in two-dimensional planes but also considerations of depth, occlusion, and camera pose, factors that substantially increase the difficulty of object localization tasks (Li et al., 2025b).

Currently, most VLMs rely primarily on large-scale image-text pairs harvested from the webs, where spatial information tends to be sparse due to the inherent lack of three-dimensional spatial annotations (Ma et al., 2024a). Moreover, even in multimodal datasets that include spatial descriptions, task designs typically remain limited to shallow spatial understanding from static viewpoints, lacking multi-dimensional, multi-perspective spatial reasoning tasks that would enable models to develop more generalizable spatial representations (Cheng et al., 2024; Zha et al., 2025). We therefore hypothesize that VLMs' deficiencies in cross-viewpoint spatial understanding tasks stem from structural limitations in their training data.

To address this research gap, we introduce ViewSpatial-Bench, the most comprehensive benchmark for evaluating spatial localization from both camera and human perspectives. This benchmark encompasses five distinct localization recognition tasks and is supported by a reliable automated 3D orientation annotation pipeline that generates efficient, diverse, and scalable image datasets with precise directional labels. Furthermore, we utilized this automated pipeline to produce extensive spatially annotated training data for VLMs, enhancing their perceptual reasoning capabilities for spatial relationships across multiple viewpoints.

Based on ViewSpatial-Bench, we conducted a comprehensive evaluation of multiple VLMs investigating their spatial understanding performance. Results demonstrate significant limitations in spatial localization tasks, particularly when reasoning across different viewpoints. To address these limitations, we introduced well-annotated spatial data for VLM training, enabling more concrete multi-perspective spatial understanding and yielding the Multi-View Spatial Model. This approach significantly improved spatial perception across viewpoints, partially validating our hypothesis. In summary, our contributions are:

- We propose *ViewSpatial-Bench*, the most comprehensive benchmark for evaluating multi-viewpoint spatial localization across 5,700 curated samples and five task types. This benchmark systematically assesses VLMs' spatial reasoning from both camera and human perspectives, addressing a critical gap in cross-viewpoint evaluation frameworks;

- We design an automated 3D spatial annotation pipeline that efficiently generates large-scale, precisely annotated multi-view datasets. This pipeline provides rich spatial relationship data for

VLM training through automated orientation annotation, establishing important foundations for future research;

- We develop the *Multi-View Spatial Model* trained on our large-scale multi-viewpoint VQA dataset. Through systematic evaluation, we identify fundamental limitations in current models' perspective-based spatial reasoning, particularly in 3D embodied environments. Our model achieves 46.24% improvement over baselines, demonstrating our methodology's effectiveness.

## 2 RELATED WORKS

**Spatial Reasoning with VLMs.**   Recently, VLMs have demonstrated significant advancements in understanding and reasoning about visual content (Bordes et al., 2024; Deng et al., 2025). Both proprietary and open-source models have achieved impressive performance in visual question answering, image captioning, and complex multimodal reasoning tasks. These models typically incorporate image encoders and vision-language fusion modules (Cho et al., 2023; Li et al., 2023; Liu et al., 2025), pre-trained on large-scale image-text pairs (Zang et al., 2024).

However, despite current VLMs' exceptional performance on certain visual reasoning tasks, their spatial understanding capabilities remain fundamentally limited (Cheng et al., 2024; Shiri et al., 2024). When handling tasks involving spatial relationships, object localization, or embodied interaction reasoning, models typically rely on camera-centric reference frames, with their spatial understanding strictly bound to the observational viewpoint (Shiri et al., 2024; Yang et al., 2025a). This constraint impairs their generalization capabilities and practical utility in tasks requiring perspective transformation or third-person spatial comprehension, making the development of models with stronger perspective-taking awareness a critical challenge for advancing multimodal intelligence.

**Benchmarks fo Spatial Perspective-Taking.**   Several benchmarks have been proposed to evaluate spatial reasoning capabilities in VLMs, but most focus primarily on single-perspective spatial understanding. For instance, EmbSpatial-Bench (Du et al., 2024) and What'sUP (Kamath et al., 2023) concentrate on assessing models' abilities to recognize spatial relationships between objects in two-dimensional images, while VSI-Bench (Yang et al., 2025a) tests model performance on compositional visual reasoning tasks involving spatial queries. Additionally, some research explores spatial reasoning in embodied AI, such as navigation and object localization tasks, but these works predominantly rely on the agent's egocentric perspective (Song et al., 2024).

Although some benchmarks have begun to address cross-viewpoint spatial understanding, such as 3DSRBench (Ma et al., 2024b) and SPHERE (Zhang et al., 2024a), they remain insufficient in terms of multi-task comprehensiveness and depth of perspective transformation assessment.

## 3 VIEWSPATIAL-BENCH

### 3.1 OVERVIEW

We introduce ViewSpatial-Bench to quantitatively evaluate VLMs' spatial localization capabilities in 3D environments from multiple perspectives. Our benchmark contains over 5,700 question-answer pairs spanning more than 1,000 unique 3D scenes, with source imagery from the validation sets of ScanNet (Dai et al., 2017) and MS-CoCo (Lin et al., 2014). Following a construction pipeline illustrated in Figure 2, we first acquired images with complete spatial information, created metadata using existing annotations, extracted spatial relationships for specific tasks, and finally constructed and filtered the QA dataset.

ViewSpatial-Bench comprises five localization recognition tasks across two complementary perspective frameworks. From the camera perspective: (1) Object Relative Direction recognition(Cam-Rel. Dir.), which determines spatial relationships between objects directly from images; (2) Object View Orientation recognition(Cam-Obj. Ori.), which identifies the gaze direction of individuals relative to the camera from an egocentric viewpoint. These tasks evaluate VLMs' intuitive, egocentric spatial understanding abilities. From the human perspective: (3) Object Relative Direction recognition(Per-Rel. Dir.), which involves adopting the viewpoint of a character in the image to determine the spatial relationships of other objects from their perspective; (4) Object View Orientation recognition(Per-Obj.

Ori.), which requires assuming the position of a character in the image to determine the direction of their gaze; (5) Scene Simulation Relative Direction recognition(Per-Sce. Sim.), which requires modeling oneself within a spatial scene across sequential frames to determine relative positions of other objects. These latter three tasks assess VLMs' abstract, perception-dependent spatial awareness while accommodating complex human pose variations and spatial information in embodied scenarios.

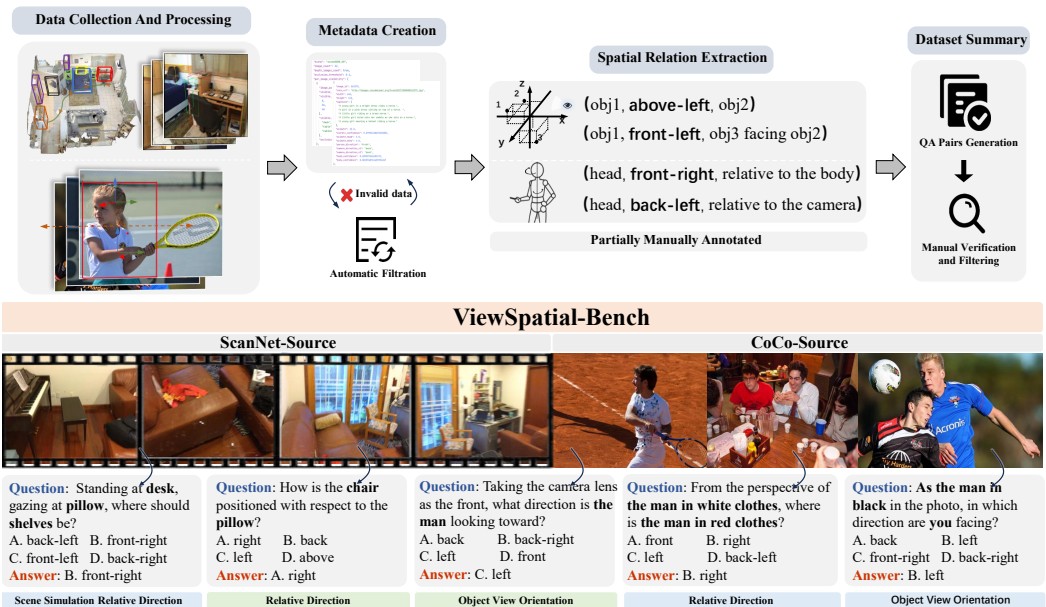

Figure 2: ViewSpatial-Bench construction pipeline. From data collection to QA generation across camera perspective ( ) and person perspective ( ) tasks. The pipeline includes metadata creation, automatic filtering, spatial relation extraction, and manual verification.

### 3.2 DATASET CONSTRUCTION

ViewSpatial-Bench construction follows a systematic process using two complementary data sources: ScanNet for rich 3D scene reconstructions with accurate spatial coordinates, and MS-CoCo for diverse images with human subjects and annotated keypoints. This combination supports both precise 3D spatial reasoning and perspective-dependent person-centric understanding tasks. We developed specialized processing pipelines for each source to extract reliable spatial relationships using automated techniques with manual verification.

**ScanNet Source.** For Cam-Rel. Dir. and Per-Sce. Sim. tasks, we utilized the ScanNet validation set. We first obtained voxel information for each scene, then applied Maximum Coverage Sampling (Algorithm 1 (Zheng et al., 2025)) to ensure complete spatial representations with minimal frames while maximizing diversity. This approach prevented redundant capture of the same spatial locations. For each selected frame, we generated scene metadata including visible objects with visibility rates and 3D spatial coordinates in the camera coordinate system.

For Cam-Rel. Dir. task, we leveraged 3D spatial coordinates and camera parameters to determine relative positions between object pairs. For Per-Sce. Sim. task, we first identified objects appearing only once in each scene (set $\mathcal{N}$), selected object triads $o_1, o_2, o_3$ from $\mathcal{N}$, and used metadata to locate frames containing all three objects. By simulating the position and orientation at $o_1$, we calculated the relative position of $o_3$ from this simulated viewpoint.

**MS-CoCo Source.** For Cam-Obj. Ori. and Per-Obj. Ori. tasks, plus Per-Rel. Dir. task, we utilized the MS-CoCo validation set. We filtered images containing animate objects occupying at least 20% of the image area.

For orientation tasks, we selected images where subjects' gaze directions aligned with head orientations. Using MS-CoCo's bounding boxes and keypoints, we segmented person images into head and body components, then employed Orient-Anything-Large (Wang et al., 2024) to calculate

rotation angles (Algorithm 2). For person-perspective orientation, we derived gaze direction by analyzing angular offsets between head and body orientations. For camera-perspective orientation, we calculated both head and body rotation angles, selecting the computation with highest confidence. For complex cases with multiple subjects, we resorted to manual annotation.

For Per-Rel. Dir. task, which include questions like "From person A's perspective, where is person B located?", we manually annotated 864 instances due to the complexity of human and object appearances and insufficient accuracy in automated approaches.

---

**Algorithm 1** Maximum Coverage Sampling

**Require:** Set of frames $F = \{f_1, f_2, \ldots, f_n\}$, voxel sets $V_k$ for each frame $f_k$, budget $K$
**Ensure:** Subset $S \subseteq F$ maximizing voxel coverage
1: Initialize $S \leftarrow \emptyset$
2: Initialize $U \leftarrow \emptyset$ {Covered voxels set}
3: **while** size of $S$ is less than $K$ **do**
4:     Select $f^* = \arg\max_{f_k \in F \setminus S} |V_k \setminus U|$
5:     Add $f^*$ to $S$
6:     Update $U \leftarrow U \cup V_{f^*}$
7:     **if** Stop condition is met **then**
8:         **break**
9:     **end if**
10: **end while**
11: **return** $S$

---

**Algorithm 2** Head-to-body Orientation Offset

**Require:** Image $I$, keypoints $K$, bounding box $B$, Orient-Anything model $D$
**Ensure:** Person gaze direction
1: $P \leftarrow \text{Crop}(I, B)$
2: $(L_x, L_y), (R_x, R_y) \leftarrow \text{ExtractShoulders}(K)$
3: **if** $\text{Visibility}(L_y) = 0$ OR $\text{Visibility}(R_y) = 0$ **then**
4:     **return** False
5: **end if**
6: $H \leftarrow \min(L_y, R_y)$
7: $P_{head} \leftarrow P[0 : H, :], P_{body} \leftarrow P[H :, :]$
8: $(az_{head}, conf_{head}) \leftarrow D(P_{head})$
9: $(az_{body}, conf_{body}) \leftarrow D(P_{body})$
10: $\Delta \leftarrow (az_{head} - az_{body} + 540) \bmod 360 - 180$
11: **return** direction based on $\Delta$ thresholds for left, front-left, front, front-right, right

---

**QA Dataset Creation.** ViewSpatial-Bench is structured as a multiple-choice benchmark derived systematically from our metadata. After extracting 3D spatial information through our ScanNet and MS-COCO processing pipelines, we converted the raw spatial coordinates and orientation angles into standardized directional relationships using a rule-based mapping system. For each task category, we designed question templates that explicitly test perspective transformation abilities. The construction followed three key steps:

First, we converted raw spatial data (3D coordinates, orientation angles) into standardized directional relationships using angle-based mapping (e.g., $22.5°$ to $67.5°$ as "front-right," $67.5°$ to $112.5°$ as "right"). This discretization enabled consistent labeling across different scenes.

Second, we populated templates with object identifiers and computed spatial relationships from our metadata. For complex spatial reasoning tasks, our templates incorporate three objects to test perspective adoption with relative positioning:

> **QA Generation Example**
>
> **Template:** "If you stand at object1 facing object2, where is object3?"
>
> **Metadata:** bookshelf(1.2, 0.5, 0), window(1.2,3.5,0), sofa(3.2,1.5,0)
>
> **Computation:**
> 1. Vector bookshelf→window: (0,3.0,0) [front direction]
> 2. Vector bookshelf→sofa: (2.0,1.0,0)
> 3. Angle: $63.43°$ clockwise = "front-right"
>
> **Question:** "If you stand at the bookshelf facing the window, where is the sofa?"
> **Answer:** "front-right"
> **Distractors:** "left", "back", "front-left"

Finally, we implemented specific rules for distractor generation: for single-directional attributes (e.g., "front"), distractors exclude compound directions containing that attribute ("front-left"); for compound directions (e.g., "front-left"), distractors exclude constituent single directions ("front" or "left"). This design systematically eliminates ambiguity and provides focused assessment of fundamental spatial concepts while controlling for question difficulty.

**Filtering and Human Verification.** To ensure the quality of ViewSpatial-Bench, we implemented a multi-stage filtering process for all tasks. During metadata generation, we eliminated invalid data with incorrectly calculated orientation angles or excessively large rotation angles. In the manual filtering stage, for relative direction tasks, we removed instances where objects were too close to each other, objects were difficult to identify, or images were blurry. For gaze direction recognition tasks, we filtered out data where subjects' gaze directions significantly differed from their head orientations or where subjects were difficult to identify. Following automated construction and filtering, we conducted manual verification to confirm that target objects were clearly visible in images and that the spatial localizations were correct and unambiguous. This iterative refinement process continued until ViewSpatial-Bench met our quality standards (Yang et al., 2025a; Du et al., 2024). Comprehensive details regarding dataset construction procedures and human annotation protocols are provided in Appendix B.1.

## 3.3 DATASET STATISTICS

Figure 3 illustrates the five task categories in ViewSpatial-Bench and their respective proportions. To ensure balanced evaluation across viewpoints, we constructed approximately equal amounts of data for camera-perspective (48.4%) and human-perspective (51.6%) tasks. This balanced distribution enables fair comparison of spatial reasoning capabilities from different observational frameworks. For the Relative Direction recognition task from camera viewpoints, which more directly demonstrates 3D scene understanding, we developed additional data to enrich spatial information diversity.

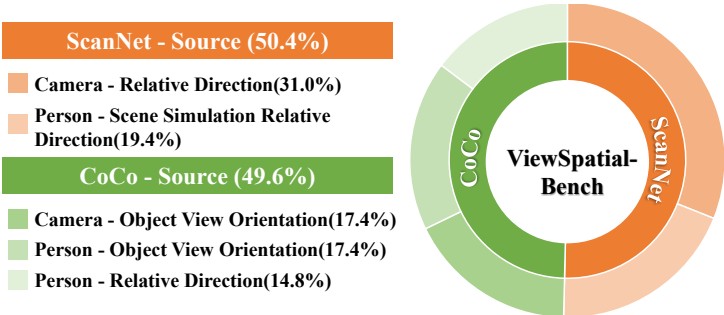

Figure 3: Distribution of task categories in ViewSpatial-Bench, balanced between ScanNet-Source and CoCo-Source approaches, with five distinct subtasks for comprehensive evaluation of spatial reasoning across different viewpoints.

Table 1 presents a comprehensive comparison between ViewSpatial-Bench and existing spatial reasoning benchmarks. ViewSpatial-Bench contains 5,712 samples across 1,338 unique scenes, employing a hybrid construction method that combines automated 3D annotation pipelines with manual verification. The benchmark distinguishes itself with 18 distinct directional categories and precise 3D coordinate annotations from ScanNet, while uniquely supporting evaluation across both camera-perspective and human-perspective viewpoints for comprehensive assessment of perspective-taking capabilities essential for embodied AI applications. Detailed statistical analysis is provided in Appendix B.2.

Table 1: Comparison of ViewSpatial-Bench with existing spatial reasoning benchmarks. ViewSpatial-Bench provides the first comprehensive evaluation framework for multi-perspective spatial localization, uniquely supporting both camera and person viewpoints with the broadest scope of directional categories and spatial query targets.

| Benchmark | Construct Method | Visual Diversity | Scale & Diversity | | 3D Annotation | | Multi-Perspective | | Spatial Query Target | |
| --- | --- | --- | --- | --- | --- | --- | --- | --- | --- | --- |
| | | | Samples | Scenes | Directions | 3D-Coord | Camera | Person | Person-Target | Object-Target |
| SpatialRGPT-Bench (Cheng et al., 2024) | Automated | Single | 1,410 | 524 | 6 | ✓ | ✓ | ✗ | ✗ | ✓ |
| EmbSpatial-Bench (Du et al., 2024) | Automated | Single | 3,640 | 284 | 6 | ✓ | ✓ | ✗ | ✗ | ✓ |
| What'sUP (Kamath et al., 2023) | Manual | Single | 820 | 205 | 12 | ✗ | ✓ | ✗ | ✗ | ✓ |
| VSI-Bench (Yang et al., 2025a) | Automated | Multi | 3,672 | 245 | 8 | ✓ | ✓ | ✓ | ✗ | ✓ |
| 3DSRBench (Ma et al., 2024b) | Manual | Single | 2,772 | 1,827 | 8 | ✗ | ✓ | ✗ | ✓ | ✓ |
| SPHERE (Zhang et al., 2024a) | Manual | Single | 2,285 | 1,001 | 7 | ✗ | ✓ | ✓ | ✓ | ✓ |
| All-Angles Bench (Yeh et al., 2025) | Hybrid | Single | 2,132 | 90 | 4 | ✗ | ✓ | ✗ | ✓ | ✓ |
| GSR-BENCH (Rajabi & Kosecka, 2024) | Automated | Multi | 820 | 205 | 12 | ✗ | ✓ | ✗ | ✗ | ✓ |
| **ViewSpatial-Bench** | **Hybrid** | **Multi** | **5,712** | **1,338** | **18** | ✓ | ✓ | ✓ | ✓ | ✓ |

## 4 MULTI-VIEW SPATIAL FINE-TUNING

To address the limitations in perspective-dependent spatial reasoning identified in current VLMs, we fine-tune existing models on our multi-perspective spatial dataset. For convenience, we refer to the resulting fine-tuned models as Multi-View Spatial Models (MVSM). Following the ViewSpatial-Bench construction pipeline, we leveraged our automated spatial annotation framework to generate approximately 43K diverse spatial relationship samples across all five task categories. This dataset incorporates 3D spatial information from ScanNet (Dai et al., 2017) and MS-COCO (Lin et al., 2014) training sets, supplemented with Spatial-MM (Shiri et al., 2024) data for the Person-perspective Relative Direction task where full automation proved challenging due to complex human spatial coordinates and environmental contexts. We structured all training data as image-text pairs using consistent natural language templates to articulate spatial relationships between objects or entities, with answers represented as standardized directional classifications. Our fine-tuning strategy trains models on mixed batches containing samples from both camera and human perspectives, encouraging the development of unified representations of 3D spatial relationships that support robust reasoning across different viewpoints.

## 5 EXPERIMENTS

### 5.1 EXPERIMENTAL SETUP

**Baselines and Metrics.** We conducted comprehensive evaluations of current VLMs on ViewSpatial-Bench using accuracy as our primary metric. Our evaluation includes a diverse set of models spanning different architectures and parameter scales: (1) Open-source models: InternVL2.5/VL3 (Chen et al., 2024b; Zhu et al., 2025), LLaVA-NeXT-Video (Zhang et al., 2024b), LLaVA-OneVision (Li et al., 2024), Llama-3.2-Vision (Grattafiori et al., 2024), Kimi-VL-Instruct (Team et al., 2025), and Qwen2.5-VL (Bai et al., 2025); (2) Proprietary models: GPT-4o (Hurst et al., 2024), GPT-5 (OpenAI, 2025), Gemini-2.0-Flash (Team et al., 2024), and Gemini-2.5-pro (Team et al., 2024); (3) Specialized spatial reasoning models: VST-3B-SFT (Yang et al., 2025b), VST-7B-SFT (Yang et al., 2025b), MindCube-3B-RawQA-SFT (Yin et al., 2025), SpatialLadder-3B (Li et al., 2025a), and SpaceQwen2.5-VL-3B (Jia et al., 2025).

**Implementation Details.** We employ our MVSM fine-tuning strategy on Qwen2.5-VL-3B (Bai et al., 2025) as the backbone model. Detailed training configurations and evaluation procedures are provided in Appendix C.1 and C.2.

### 5.2 MAIN RESULTS

As shown in Table 2, our comprehensive evaluation reveals critical insights into the spatial reasoning capabilities of current VLMs and validates our approach:

**Fundamental limitations in perspective-based spatial reasoning:** Even powerful proprietary models like GPT-4o (34.98%) and Gemini-2.0-Flash (32.56%) demonstrate surprisingly weak spatial localization capabilities, barely outperforming random chance (26.33%). This confirms our hypothesis presented in the introduction that current VLMs, despite their impressive performance on standard vision-language tasks, fundamentally struggle with perspective-dependent spatial reasoning. The consistently poor performance across diverse architectures suggests this is not merely an implementation issue but a systematic deficiency in how these models conceptualize spatial relationships.

**Egocentric vs. allocentric reasoning gap:** Most VLMs exhibit an intriguing pattern wherein their spatial localization accuracy from camera perspectives (averaging 35.5%) falls below their performance from human viewpoints (averaging 36.9%). This contradicts the intuitive expectation that egocentric perspective (camera-based) should be easier than allocentric reasoning (human-based). This finding aligns with our observation that VLMs lack the perspective-taking ability that humans naturally possess, and suggests that current vision-language architectures may implicitly encode certain spatial biases that favor third-person viewpoints, potentially due to the prevalence of such compositions in web-harvested training data.

**Task-specific performance asymmetries:** A particularly revealing pattern emerges in the interaction between task type and perspective. Most VLMs perform significantly worse on Object View Orien-

Table 2: Zero-shot performance on ViewSpatial-Bench. Accuracy comparison across multiple VLMs on camera and human perspective spatial tasks. Our Multi-View Spatial Model (MVSM) significantly outperforms all baseline models across all task categories, demonstrating the effectiveness of our multi-perspective spatial fine-tuning approach.

| Model | Camera-based Tasks | | | Person-based Tasks | | | | Overall |
|---|---|---|---|---|---|---|---|---|
| | Rel. Dir. | Obj. Ori. | Avg. | Obj. Ori. | Rel. Dir. | Sce. Sim. | Avg. | |
| *Proprietary Models* | | | | | | | | |
| GPT-4o(Hurst et al., 2024) | 41.46 | 19.58 | 33.57 | 42.97 | 40.86 | 26.79 | 36.29 | 34.98 |
| Gemini-2.0-Flash (Team et al., 2024) | 45.29 | 12.95 | 33.66 | 41.16 | 32.78 | 21.90 | 31.53 | 32.56 |
| GPT-5 (OpenAI, 2025) | 53.52 | **25.60** | 43.48 | 58.23 | 46.32 | 38.46 | 47.40 | 45.49 |
| Gemini-2.5-pro (Team et al., 2024) | **54.30** | 24.30 | **43.52** | **59.84** | **47.27** | **39.28** | **48.52** | **46.09** |
| *Open-Source General Models* | | | | | | | | |
| InternVL2.5 (2B) (Chen et al., 2024b) | 38.52 | 22.59 | 32.79 | 47.09 | 40.02 | 25.70 | 37.04 | 34.98 |
| Qwen2.5-VL (7B) (Bai et al., 2025) | 46.64 | 29.72 | 40.56 | 37.05 | 35.04 | 28.78 | 33.37 | 36.85 |
| LLaVA-NeXT-Video (7B) (Zhang et al., 2024b) | 26.34 | 19.28 | 23.80 | 44.68 | 38.60 | 29.05 | 37.07 | 30.64 |
| LLaVA-OneVision (7B) (Li et al., 2024) | 29.84 | 26.10 | 28.49 | 22.39 | 31.00 | 26.88 | 26.54 | 27.49 |
| InternVL2.5 (8B) (Chen et al., 2024b) | 49.41 | **41.27** | 46.48 | 46.79 | 42.04 | **32.85** | 40.20 | **43.24** |
| Llama-3.2-Vision (11B) (Grattafiori et al., 2024) | 25.27 | 20.98 | 23.73 | 51.20 | 32.19 | 18.82 | 33.61 | 28.82 |
| InternVL3 (14B) (Zhu et al., 2025) | **54.65** | 33.63 | **47.09** | 33.43 | 37.05 | 31.86 | 33.88 | 40.28 |
| Kimi-VL-Instruct (16B) (Team et al., 2025) | 26.85 | 22.09 | 25.14 | **63.05** | **43.94** | 20.27 | **41.52** | 33.58 |
| *Multi-View Spatial Fine-Tuning* | | | | | | | | |
| Qwen2.5-VL (3B) (Bai et al., 2025) [Backbone] | 43.43 | 33.33 | 39.80 | 39.16 | 28.62 | 28.51 | 32.14 | 35.85 |
| **Multi-View Spatial Model** | **83.59** | **87.65** | **85.05** | **90.16** | **71.14** | **75.75** | **79.31** | **82.09** |
| *Improvement over backbone* | +40.16 | +54.32 | +45.25 | +51.00 | +42.52 | +47.24 | +47.17 | +46.24 |
| Random Baseline | 25.16 | 26.10 | 25.50 | 24.60 | 31.12 | 26.33 | 27.12 | 26.33 |

tation tasks from camera perspectives compared to Relative Direction tasks, yet show the opposite pattern for human perspective tasks (45.2% for Object View Orientation vs. 38.1% for Relative Direction). This striking asymmetry confirms our hypothesis that current VLMs lack consistent cross-viewpoint spatial understanding. The discrepancy suggests these models fail to construct a coherent 3D representation that can be flexibly navigated from different viewpoints, instead treating different perspective-task combinations as essentially separate problems.

**Effectiveness of perspective-aware training:** Our Multi-View Spatial Model achieves dramatic improvement compared to its backbone Qwen2.5-VL (3B) model, representing a 46.24% absolute performance gain. The model shows remarkably consistent improvements across all task categories. The most substantial gains occur in orientation tasks, with improvements of 54.32% for camera-perspective and 51.00% for human-perspective Object View Orientation tasks. This symmetrical improvement pattern is particularly noteworthy, as it demonstrates that explicit training on diverse spatial annotations with perspective awareness enables the development of unified 3D spatial representations that function effectively across viewpoints.

Table 3: Performance of specialized spatial reasoning models on ViewSpatial-Bench. These models have been specifically trained with spatial-oriented datasets and dedicated training strategies, demonstrating the effectiveness of targeted spatial training approaches.

| Model | Camera-based Tasks | | | Person-based Tasks | | | | Overall |
|---|---|---|---|---|---|---|---|---|
| | Rel. Dir. | Obj. Ori. | Avg. | Obj. Ori. | Rel. Dir. | Sce. Sim. | Avg. | |
| VST-3B-SFT (Yang et al., 2025b) | 47.49 | 43.47 | 46.05 | 68.37 | 51.31 | 62.35 | **61.23** | 53.87 |
| VST-7B-SFT (Yang et al., 2025b) | **68.75** | **72.49** | **70.10** | 50.70 | **58.67** | **64.25** | 58.07 | **63.90** |
| MindCube-3B-RawQA-SFT (Yin et al., 2025) | 49.58 | 59.64 | 53.20 | 32.23 | 31.38 | 21.45 | 28.07 | 40.25 |
| SpatialLadder-3B (Li et al., 2025a) | 48.34 | 24.10 | 39.62 | **71.08** | 34.44 | 38.91 | 48.52 | 44.21 |
| SpaceQwen2.5-VL-3B (Jia et al., 2025) | 48.17 | 20.08 | 38.06 | 53.51 | 37.77 | 33.39 | 41.46 | 39.81 |
| Random Baseline | 25.16 | 26.10 | 25.50 | 24.60 | 31.12 | 26.33 | 27.12 | 26.33 |

**Specialized Spatial Reasoning Baselines:** To provide comprehensive baseline comparisons, we evaluated several models specifically designed for spatial reasoning tasks, as shown in Table 3. VST-7B-SFT achieves the strongest performance among specialized baselines at 63.90% overall accuracy, demonstrating substantial improvements over general-purpose VLMs. The consistent gains of these specialized models across both camera-based and person-based task categories validate the critical importance of high-quality spatial annotations and targeted training methodologies. These results confirm that effective perspective-aware spatial understanding requires dedicated training approaches, with performance improvements strongly dependent on the comprehensiveness and quality of the underlying spatial training data.

## 5.3 EMPOWERING SPATIAL INTERACTION APPLICATION

To further validate MVSM's spatial understanding capabilities in practical applications, we evaluated its performance on VSI-Bench (Yang et al., 2025a) in typical tasks requiring perspective transformation, including Object Relative Direction and Route Planning subtasks. Additionally, we constructed a small application evaluation dataset, ViewSpatial Interaction Application Dataset (VSI-App), encompassing both indoor and outdoor scenarios, specifically designed to assess spatial orientation recognition abilities in embodied interaction environments, with particular focus on the requirements for dynamic scene and multi-perspective understanding during human-machine interaction.

### 5.3.1 TRANSFER LEARNING PERFORMANCE

As shown in Table 4, we assessed MVSM's generalization capabilities on both VSI-Bench and our custom VSI-App benchmark. The specific construction process and evaluation methods of the VSI-App are shown in Appendix B.4.

Table 4: Performance comparison of our Multi-View Spatial Model against its backbone.

| Model | VSI-Bench | | | VSI-App | | |
|---|---|---|---|---|---|---|
| | Rel Dir | Route Plan | Average | Indoor | Outdoor | Average |
| GPT-4o (Hurst et al., 2024) | 41.30 | 31.50 | 39.66 | 34.00 | 27.00 | 30.50 |
| Qwen2.5-VL(3B) (Bai et al., 2025) | 46.00 | 21.90 | 41.97 | 18.00 | 27.00 | 22.50 |
| **MVSM** | **46.93** $^{\uparrow 0.93}$ | **31.44** $^{\uparrow 9.54}$ | **44.34** $^{\uparrow 2.37}$ | **41.00** $^{\uparrow 23.00}$ | **36.00** $^{\uparrow 9.00}$ | **38.50** $^{\uparrow 16.00}$ |

**VSI-Bench Evaluation**: We selected two representative tasks requiring perspective transformation abilities: Object Relative Direction and Route Planning. The former requires determining spatial relationships between objects in complex indoor scenes, while the latter involves inferring and completing reasonable navigation paths. MVSM outperforms its backbone model in both tasks, with particularly significant gains in Route Planning (+9.54%). This improvement demonstrates MVSM's enhanced ability to model not just static spatial relationships but also dynamic trajectories through 3D environments, which emerged from our perspective-aware training approach without explicit route planning optimization.

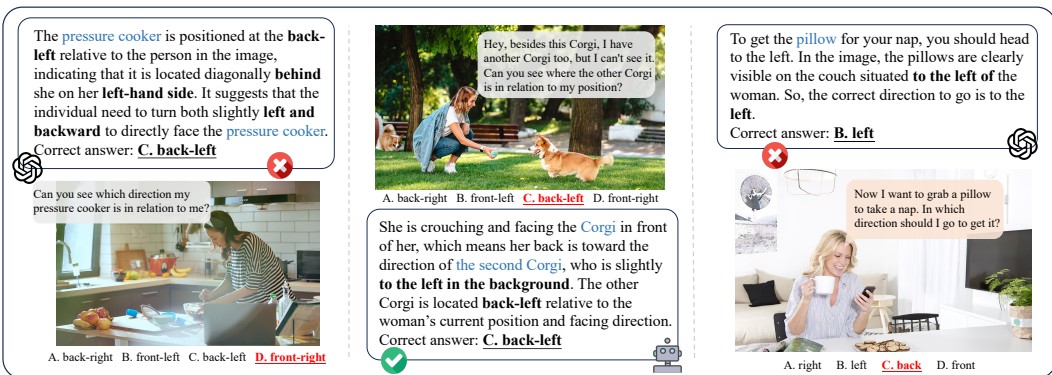

Figure 4: The image compares spatial reasoning performance between GPT-4o and MVSM on the VSI-App dataset, showing several examples where MVSM correctly answers perspective-taking questions about object locations, while GPT-4o makes errors when attempting to determine spatial relationships from another person's viewpoint.

**VSI-App Evaluation:** To further approximate real-world interaction scenarios, we constructed VSI-App, a specialized evaluation dataset of 50 scenes (25 indoor, 25 outdoor) designed to assess human-centric spatial reasoning in embodied contexts. The benchmark requires models to perform spatial reasoning from human first-person perspectives, generating responses that conform to human cognitive patterns. MVSM shows substantial improvement in indoor environments (+20.00%) and modest gains in outdoor scenarios (+4.00%). This performance pattern reveals an interesting domain gap: indoor environments with structured spatial relationships better align with our training distribution, while outdoor scenes pose greater challenges despite still showing improvement.

### 5.3.2 PERSPECTIVE CONFUSION ANALYSIS

The performance improvement on our benchmarks stems directly from MVSM's enhanced ability to maintain consistent perspective representations. To illustrate this capability, Figure 4 contrasts MVSM with GPT-4o on representative VSI-App examples requiring perspective transformation. While GPT-4o demonstrates some ability to locate objects from human perspectives, it frequently defaults to camera-centric judgments for orientation determinations, resulting in perspective confusion.

Analysis of failure modes reveals that models without perspective-aware training demonstrate inconsistent spatial judgments within single responses, alternating between human and camera perspectives. This suggests they lack a coherent internal model of 3D space that can be navigated from different viewpoints. In contrast, MVSM maintains consistent adherence to the specified perspective frame, even in challenging cases requiring multiple spatial transformations.

## 6 CONCLUSIONS

In this work, we present ViewSpatial-Bench, the first comprehensive benchmark for evaluating multi-perspective spatial localization capabilities of vision-language models across five distinct task types. Our assessment of various advanced VLMs reveals significant limitations in their spatial reasoning abilities. By developing an automated spatial annotation pipeline and constructing a large-scale multi-perspective dataset, we successfully trained our Multi-View Spatial Model (MVSM), which achieves substantial overall performance improvements on ViewSpatial-Bench tasks. Further experiments on VSI-Bench and our custom VSI-App dataset demonstrate MVSM's generalization capabilities to real-world embodied interaction scenarios. Our work establishes a foundation for spatially intelligent VLMs that better align with human cognitive patterns in embodied environments, representing an important step toward more intuitive and effective human-machine spatial communication.

### ETHICS STATEMENT

The research presented here centers on fundamental computational challenges in spatial reasoning for vision-language models using established computer vision datasets. Our work leverages publicly accessible resources including ScanNet indoor scenes, MS-COCO object annotations, and Spatial-MM spatial relationship data, all of which have been previously vetted by the research community. The automated annotation pipeline we developed operates solely on geometric and spatial metadata without processing personally identifiable information or requiring human subject participation. Our approach to multi-perspective spatial training represents a technical advancement in model capabilities rather than an application with direct societal implications. The ViewSpatial-Bench evaluation framework addresses a core limitation in current AI systems by measuring their ability to understand spatial relationships from different viewpoints, which has applications in robotics, navigation, and human-computer interaction domains that ultimately serve to improve assistive technologies and accessibility tools.

### REPRODUCIBILITY STATEMENT

Our experimental methodology has been designed with reproducibility as a primary consideration throughout the research process. The paper provides comprehensive algorithmic descriptions for our Maximum Coverage Sampling procedure and Head-to-body Orientation Offset calculation, following the same parameter configurations as their original implementations, while all novel parameters introduced in our work are explicitly detailed within the main text and Appendix B.1. Our multi-stage data construction pipeline includes detailed filtering criteria, annotation verification procedures, and quality assurance mechanisms that enable other researchers to reconstruct comparable datasets. Training procedures specify exact hyperparameters, learning rates, batch sizes, and hardware configurations used across all experiments, as detailed in Appendix C.1. The evaluation protocol documents model selection criteria, inference settings, and statistical analysis methods with sufficient granularity to ensure consistent replication, as specified in Appendix C.2. Beyond methodological transparency, we have structured our work to support the broader research community by preparing our codebase, annotation tools, and benchmark datasets for public release, thereby enabling both direct replication of our results and extension of our methods to new domains and applications.

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

## A  LIMITATIONS

While ViewSpatial-Bench represents a significant step forward in evaluating multi-perspective spatial reasoning in VLMs, several limitations merit acknowledgment:

**Annotation Challenges for Human-Perspective Tasks.**  The Person-perspective Relative Direction task presented substantial annotation challenges. The inherent complexity of human spatial coordinates and environmental contexts in natural images prevented full automation of the annotation process. This necessitated manual labeling, which introduces both scaling constraints and potential annotator biases. Future work could explore semi-supervised approaches that might reduce the reliance on manual annotation while maintaining data quality.

**Domain Constraints in Environmental Coverage.**  Our Camera-perspective Relative Direction tasks utilize exclusively indoor environments from ScanNet (Dai et al., 2017), potentially limiting generalizability to outdoor settings. As our transfer learning experiments on VSI-App suggest, there exists a substantial domain gap between indoor and outdoor spatial reasoning tasks. Outdoor environments present different spatial scales, object densities, and visual characteristics that may require specialized training approaches beyond those presented in this work.

**Static vs. Dynamic Spatial Reasoning.**  ViewSpatial-Bench evaluates only static spatial orientation comprehension without addressing dynamic spatial reasoning scenarios where objects or observers move through environments. Such dynamic reasoning represents an important aspect of embodied spatial cognition relevant to many practical applications, including robot navigation and interactive systems (Li et al., 2025c). Extending our benchmark to incorporate temporal sequences and motion-based spatial reasoning would provide a more comprehensive evaluation framework for embodied AI systems.

These limitations point to promising directions for future research that could build upon the foundation established by ViewSpatial-Bench while addressing its current constraints.

## B  DATA DETAILS

### B.1  DATASET COLLECTION AND UNIFICATION

**ScanNet Data Collection.**  We employ a three-stage video frame sampling strategy to optimize benchmark data quality: first extracting all video frames, then uniformly sampling every 10th frame, and finally applying maximum frame sampling to select the minimal yet comprehensive set of consecutive frames that capture complete scene information.

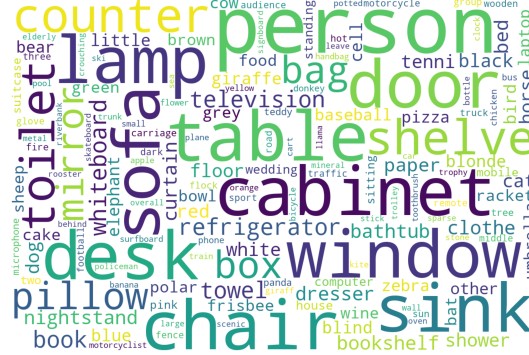

Figure 5: Wordcloud of object categories.

For 3D bounding box visibility analysis, we utilize a depth-aware projection technique that transforms 3D bounding boxes from world coordinates to camera view while accounting for occlusions. Our implementation aligns depth and color frames using scale factors (1000.0 mm to m) and handles resolution differences through proportional coordinate mapping. The occlusion detection compares the computed depth of 3D bounding box vertices against the measured depth from sensor data with a 0.1m threshold, enabling accurate determination of vertex visibility. This approach generates precise visibility annotations by requiring at least 1% of vertices to be visible for an object to be considered present in a frame, enhancing the fidelity of our object detection and 3D reasoning benchmarks.

**MS-CoCo Data Collection.**  Based on MS-CoCo (Lin et al., 2014) dataset annotations, we filter samples containing biological objects that occupy at least 20% of the image area to ensure sufficient

visual salience of target objects. We subsequently employ manual annotation to filter out samples where gaze direction significantly deviates from head orientation, ensuring consistency in spatial orientation labeling. The filtered samples are then processed by the Orient-Anything-Large model for automatic head and body orientation angle annotation. Given that this model exhibits labeling errors when processing low-resolution images or objects with ambiguous directional tendencies, we conduct focused manual verification and data correction on extreme angle samples (excessively large or small angles). This quality assurance mechanism ensures the annotation accuracy of the final dataset.

**QA Pair Generation.**  We extract object information and corresponding angle annotations from metadata for each sample. Object names are filled into predefined question templates, with computed angles serving as ground truth answers to construct multiple-choice questions. The question templates used are detailed in Table 5.

Table 5: Prompt templates used to generate spatial reasoning questions across four tasks. Object names are inserted into the templates to form natural language questions, which are later paired with direction-based multiple-choice answers derived from scene metadata.

| Task | Question Template |
|---|---|
| **Cam-Rel. Dir.** | • Can you describe the position of the {object1} relative to the {object2}?
• Could you tell me the location of the {object1} in comparison to the {object2}?
• Where is the {object1} in relation to the {object2}?
• Where is the {object1} located compared to the {object2} from the camera's perspective?
• How is the {object1} positioned with respect to the {object2}?
• If you're looking at the {object2}, where would you find the {object1}? |
| **Cam-Obj. Dir.** | • With the camera's viewpoint as the front, which direction is {object} facing in the image?
• Taking the camera lens as the front, what direction is {object} looking toward?
• Taking the camera's viewpoint as the front, which way is {object} facing in the image?
• Considering the camera's perspective as the front, what direction is {object} facing within the picture? |
| **Per-Obj. Dir.** | • Imagine you're {object} in this image — which direction are you facing?
• Suppose you are in {object}'s position, what direction are you facing?
• Picture yourself as {object}; which way are you looking in the scene?
• As {object} in the photo, in which direction are you facing? |
| **Per-Sce. Sim.** | • Imagine standing at {object1} looking towards {object2}, where is {object3}?
• When positioned at {object1} facing {object2}, where can you find {object3}?
• If you stand at {object1} facing {object2}, where is {object3}?
• Standing at {object1}, gazing at {object2}, where should {object3} be? |

**Human Annotation Protocol.**  To ensure annotation quality and reliability, we recruited five graduate-level annotators with backgrounds in computer vision and spatial reasoning. The annotation process followed a systematic verification protocol evaluating three key criteria: image clarity (whether objects and spatial relationships are visually discernible), spatial discriminability (whether directional relationships can be unambiguously determined), and scene complexity (whether the spatial configuration supports clear perspective-taking tasks). Each generated sample underwent independent review by at least two annotators who assessed whether the automatically generated spatial labels accurately reflected the visual content. Samples exhibiting ambiguous spatial relationships, poor image quality, or inconsistent annotations between reviewers were directly removed from the benchmark. This rigorous filtering process ensured that the final benchmark contains only high-quality, unambiguous spatial reasoning instances suitable for reliable model evaluation.

## B.2 DATA STATISCS

As shown in the word cloud analysis in Figure 5, our dataset is primarily constructed around two major categories: humans and objects, which aligns with our dual spatial localization task design targeting both camera and human perspectives. Table 6 provides a detailed breakdown of sample distributions across different task types in ViewSpatial-Bench.

Figure 6 shows the frequency distribution of spatial prepositions and objects in ViewSpatial-Bench. As illustrated in Figure 4(a), our benchmark incorporates a comprehensive set of directional terms, with primary directions ("front", "right", "left") showing higher frequency than compound directions ("front-left", "back-right", "above-left"). This diverse coverage ensures thorough evaluation of VLMs' ability to process complex spatial relationships from multiple perspectives, reflecting the natural usage patterns of spatial language.

Figure 6(b) depicts the distribution of the top 20 objects in ViewSpatial-Bench. The object distribution reflects common entities encountered in everyday environments, with furniture items (chair, table, sofa, desk) and personal objects well represented. This ensures practical relevance of the benchmark to real-world spatial reasoning scenarios, particularly for embodied AI applications that must navigate and interact with common objects.

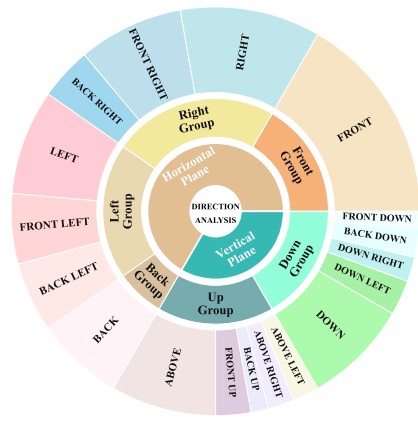

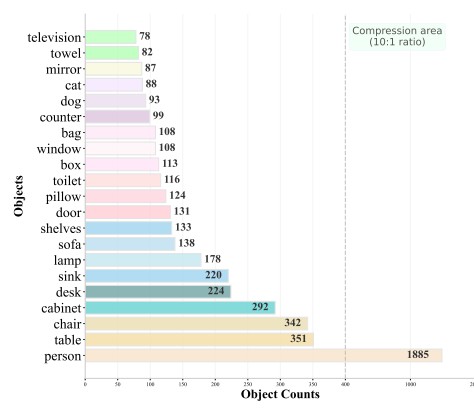

(a) Answer Direction Distribution    (b) Top 20 Objects Frequency

Figure 6: Frequency distributions in ViewSpatial-Bench. (a) Distribution of spatial prepositions, showing comprehensive coverage of directional relationships. (b) Frequency of the top 20 objects, demonstrating the benchmark's focus on common entities encountered in everyday environments.

### B.3 DATA CASES

Figures 7–9 illustrates response examples from different models across various question types in ViewSpatial-Bench.

### B.4 VSI-APP DATASET CONSTRUCTION

For the ViewSpatial Interaction Application Dataset (VSI-App), we employ a three-stage human curation approach to construct a dataset specifically designed to evaluate multi-view spatial models (MVSM) capabilities in spatial reasoning for human-computer interaction under Out-of-Distribution scenarios. Initially, two professional annotators carefully screened and downloaded 200 high-quality scene images from professional online image platforms, with 100 indoor and 100 outdoor scenes respectively. Image selection strictly adheres to the following criteria: scenes must be highly consistent with indoor/outdoor themes, contain rich three-dimensional spatial hierarchical information, include clearly identifiable human subjects as viewpoint references, and demonstrate explicit spatial relationships and potential interaction possibilities between humans and other objects in the scene. This meticulous scene selection ensures that the dataset can adequately simulate the complex spatial environments of real-world human-computer interactions.

In the question annotation phase, two annotators conduct in-depth spatial analysis of the primary human subjects in each image, focusing on two core interaction scenarios: first, spatial cognition questions where human subjects inquire about the relative positions of other objects from their first-person perspective, and second, path planning and navigation orientation questions from the human's current position to target locations. The annotators completely abandon template-based QA generation methods, directly employing natural language that closely resembles daily communication

for question descriptions, while meticulously designing accurate ground truth answers and plausible distractors for each question. This natural language annotation approach not only enhances question diversity and authenticity, but more importantly captures the linguistic expression habits and cognitive patterns of humans in actual spatial interactions.

VSI-App aims to verify whether MVSM can accurately understand and respond to spatial reasoning inquiries from human perspectives when confronted with realistic human-computer interaction scenarios, thereby evaluating the model's generalization capability and practical utility. Evaluation follows a multiple-choice format, with specific examples shown in Figure 4.

Table 6: Sample counts for different tasks in ViewSpatial-Bench evaluation and MVSM training data.

| | Camera | | | Person | | | | Overall |
|---|---|---|---|---|---|---|---|---|
| | Rel. Dir. | Obj. Dir. | Sum. | Obj. Dir. | Rel. Dir. | Sce. Sim. | Sum. | |
| **Test** | 1773 | 996 | 2769 | 996 | 842 | 1105 | 2943 | **5712** |
| **Train** | 13644 | 8954 | 22598 | 8954 | 1014 | 10309 | 20277 | **42875** |

## C  EXPERIMENTS

### C.1  IMPLEMENTATION DETAILS

We select Qwen2.5-VL-3B (Bai et al., 2025) as the base model for supervised fine-tuning. The Cam-Rel. Dir., Cam-Obj. Ori., Per-Obj. Ori., and Per-Sce. Sim. tasks in the training dataset are generated through our automated construction pipeline using unified QA templates. The Per-Rel. Dir. task is constructed based on the Spatial-MM Shiri et al. (2024) dataset, with language models employed to polish questions and enhance sample diversity. The distribution of training samples across tasks is detailed in Table 6.

Following standard practice in efficient adaptation, we freeze the vision encoder and multi-modal projector while keeping the language model trainable. The model is trained for 3 epoch with an effective batch size of 16, achieved through gradient accumulation (4 steps with per-device batch size of 1) across 4 NVIDIA A100 (40GB) GPUs. The entire training process requires approximately 8.5 GPU hours, making our approach computationally efficient and accessible.

### C.2  EVALUATION DETAILS

**ViewSpatial-Bench evaluation.**  We evaluate all models under zero-shot settings, where models must directly predict the correct option based on given images and questions. For API-based models, we used their standard online interfaces with default parameters. For open-source models, we employed their default generation settings through the Transformers (Wolf et al., 2020) library. To ensure consistency and reliability in our results, each model was evaluated five times. The results reported in the manuscript represent the average performance across these multiple runs. Accuracy is calculated by comparing model predictions with ground truth answers. The prompt template used for evaluation is shown below.

> **Zero-shot Evaluation Prompt**
>
> Question:{question}
> Choices:{choices}
> Reply only to the corresponding option.
> Answer:

**VSI-Bench evaluation.**  We follow the original paper's experimental settings for VSI-Bench (Yang et al., 2025a) evaluation. We employ the lmms-eval framework to conduct zero-shot testing with a batch size of 1 and maximum frame count set to 32. All models are evaluated on a single GPU environment (A6000 48G) using the accelerate launcher.

**VSI-App dataset evaluation.** Since VSI-App is a small-scale test benchmark designed for Out-of-Distribution scenarios, we adopt a repeated testing strategy to enhance evaluation reliability. Specifically, we generate 5 different option orderings for each question sample and conduct 5 independent tests for each model on these reordered samples. The final answer is determined through a voting mechanism, selecting the option with the highest frequency across the 5 tests for the same question as the prediction result. This method effectively reduces the potential impact of option ordering on model predictions.

**Statistical Robustness Analysis** As described in the evaluation protocol above, we conducted five independent evaluation runs for all models on both ViewSpatial-Bench and VSI-App to ensure result reliability. Table 7 reports the mean accuracy with standard deviations (shown as subscripts). The low standard deviations across all models indicate that the observed results are stable and reproducible.

Table 7: Statistical robustness analysis. Results are reported as mean with standard deviation over 5 independent runs.

| Model | ViewSpatial-Bench | | | VSI-App | | |
|---|---|---|---|---|---|---|
| | Overall | Camera-Avg | Person-Avg | Average | Indoor | Outdoor |
| *Proprietary Models* | | | | | | |
| GPT-4o | $34.98_{\pm 0.53}$ | $33.57_{\pm 0.55}$ | $36.29_{\pm 0.58}$ | $30.50_{\pm 0.55}$ | $34.00_{\pm 0.71}$ | $27.00_{\pm 0.89}$ |
| Gemini-2.0-Flash | $32.56_{\pm 0.47}$ | $33.66_{\pm 0.49}$ | $31.53_{\pm 0.45}$ | - | - | - |
| GPT-5 | $45.49_{\pm 0.61}$ | $43.48_{\pm 0.59}$ | $47.40_{\pm 0.64}$ | - | - | - |
| Gemini-2.5-pro | $46.09_{\pm 0.58}$ | $43.52_{\pm 0.56}$ | $48.52_{\pm 0.60}$ | - | - | - |
| *Open-Source General Models* | | | | | | |
| InternVL2.5 (2B) | $34.98_{\pm 0.49}$ | $32.79_{\pm 0.47}$ | $37.04_{\pm 0.51}$ | - | - | - |
| Qwen2.5-VL (7B) | $36.85_{\pm 0.55}$ | $40.56_{\pm 0.57}$ | $33.37_{\pm 0.52}$ | - | - | - |
| LLaVA-NeXT-Video (7B) | $30.64_{\pm 0.64}$ | $23.80_{\pm 0.61}$ | $37.07_{\pm 0.66}$ | - | - | - |
| LLaVA-OneVision (7B) | $27.49_{\pm 0.52}$ | $28.49_{\pm 0.54}$ | $26.54_{\pm 0.50}$ | - | - | - |
| InternVL2.5 (8B) | $43.24_{\pm 0.46}$ | $46.48_{\pm 0.48}$ | $40.20_{\pm 0.44}$ | - | - | - |
| Llama-3.2-Vision (11B) | $28.82_{\pm 0.59}$ | $23.73_{\pm 0.56}$ | $33.61_{\pm 0.61}$ | - | - | - |
| InternVL3 (14B) | $40.28_{\pm 0.51}$ | $47.09_{\pm 0.53}$ | $33.88_{\pm 0.48}$ | - | - | - |
| Kimi-VL-Instruct (16B) | $33.58_{\pm 0.48}$ | $25.14_{\pm 0.46}$ | $41.52_{\pm 0.50}$ | - | - | - |
| *Specialized Spatial Reasoning Models* | | | | | | |
| VST-3B-SFT | $53.87_{\pm 0.56}$ | $46.05_{\pm 0.53}$ | $61.23_{\pm 0.58}$ | - | - | - |
| VST-7B-SFT | $63.90_{\pm 0.62}$ | $70.10_{\pm 0.65}$ | $58.07_{\pm 0.59}$ | - | - | - |
| MindCube-3B-RawQA-SFT | $40.25_{\pm 0.51}$ | $53.20_{\pm 0.54}$ | $28.07_{\pm 0.48}$ | - | - | - |
| SpatialLadder-3B | $44.21_{\pm 0.48}$ | $39.62_{\pm 0.46}$ | $48.52_{\pm 0.50}$ | - | - | - |
| SpaceQwen2.5-VL-3B | $39.81_{\pm 0.53}$ | $38.06_{\pm 0.51}$ | $41.46_{\pm 0.55}$ | - | - | - |
| *Multi-View Spatial Fine-Tuning* | | | | | | |
| Qwen2.5-VL (3B) | $35.85_{\pm 0.57}$ | $39.80_{\pm 0.59}$ | $32.14_{\pm 0.55}$ | $22.50_{\pm 0.55}$ | $18.00_{\pm 0.45}$ | $27.00_{\pm 0.71}$ |
| Multi-View Spatial Model | $82.09_{\pm 0.54}$ | $85.05_{\pm 0.56}$ | $79.31_{\pm 0.52}$ | $38.50_{\pm 0.55}$ | $41.00_{\pm 0.71}$ | $36.00_{\pm 0.45}$ |
| Random Baseline | $26.33_{\pm 0.31}$ | $25.50_{\pm 0.28}$ | $27.12_{\pm 0.33}$ | - | - | - |

## C.3 ANALYSIS EXPERIMENT

**Training Format and Shortcut Learning Analysis.** To verify that our Multi-View Spatial Model's performance improvements stem from genuine spatial reasoning rather than shortcut learning, we conducted controlled experiments comparing different training formats. We used the same multi-perspective spatial dataset described in Section 3.3, containing approximately 43K samples across all five task categories. To eliminate potential shortcut learning through option elimination strategies, we converted the original multiple-choice format into a direct answer format where models generate spatial directions without candidate options.

Following identical experimental settings as described in Section 5.1, we trained Qwen2.5-VL(3B) on both formats. As shown in table 8, both approaches yielded substantial improvements over the baseline, with multiple-choice format achieving 82.09% overall accuracy compared to 79.34% for direct answer format. The minimal performance difference between formats confirms that MVSM's gains result from enhanced spatial understanding rather than exploitation of multiple-choice structural patterns, validating that our approach teaches robust spatial reasoning capabilities that generalize across different response formats.

**Multi-Backbone Generalization Validation.** To demonstrate the broader applicability of our training methodology, we evaluated MVSM training procedures across multiple vision-language model architectures using the same multi-perspective spatial dataset described in Section 3.3 and the training configurations specified in Section 5.1. As shown in the table 8, results across three representative backbones show consistent substantial improvements: Qwen2.5-VL(3B) improved from 35.85% to 82.09% (+46.24%), InternVL-2B improved from 34.98% to 76.45% (+41.47%), and Qwen2.5-VL(7B) improved from 36.85% to 83.01% (+46.16%). The consistent performance gains across different model families and parameter scales demonstrate the architecture-agnostic effectiveness of our perspective-aware training approach. These results establish that the benefits of multi-perspective spatial training extend beyond specific model implementations, indicating robust transferability of our methodology across diverse vision-language architectures.

Table 8: Comprehensive analysis of MVSM training robustness across different question formats and model architectures. MC denotes Multiple Choice format, DA denotes Direct Answer format.

| Experiment | Backbone Model | Original | MVSM | Improvement |
|---|---|---|---|---|
| **Training Format** | Qwen2.5-VL (3B) | 35.85% | **82.09**% *(MC)* | +46.24% |
| | | | **79.34**% *(DA)* | +43.49% |
| **Multi-Backbone** | Qwen2.5-VL (3B) | 35.85% | **82.09%** | +46.24% |
| | Qwen2.5-VL (7B) | 36.85% | **83.01%** | +46.16% |
| | InternVL2.5 (2B) | 34.98% | **76.45%** | +41.47% |

# LLM USAGE

We acknowledge the use of large language models exclusively for writing assistance and linguistic refinement in preparing this manuscript. These tools enhanced clarity, grammatical accuracy, and academic style while preserving all original research contributions, methodological approaches, and scientific insights developed by the authors. The language models served solely as writing aids to improve sentence structure, readability of technical content, academic terminology refinement, and writing style consistency throughout the manuscript. Importantly, LLMs were not employed for research ideation, conceptual development, literature review, citation discovery, data analysis, experimental design, or generation of research hypotheses and conclusions. All research ideas, experimental work, data analysis, and scientific conclusions presented originate entirely from the authors' independent intellectual work. The authors take full responsibility for all content, including any text refined with LLM assistance, ensuring that the core intellectual contributions and scientific merit remain wholly attributable to the listed authors.

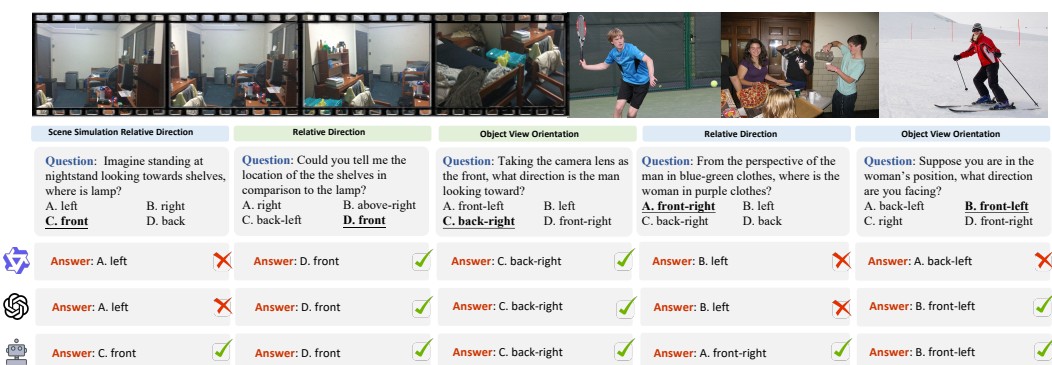

Figure 7: ViewSpatial-Bench Examples (Part1). Performance comparison of three models (Qwen2.5-VL(3B), GPT-4o, and MVSM) on five spatial reasoning tasks from camera perspective ( ) and person perspective ( ).

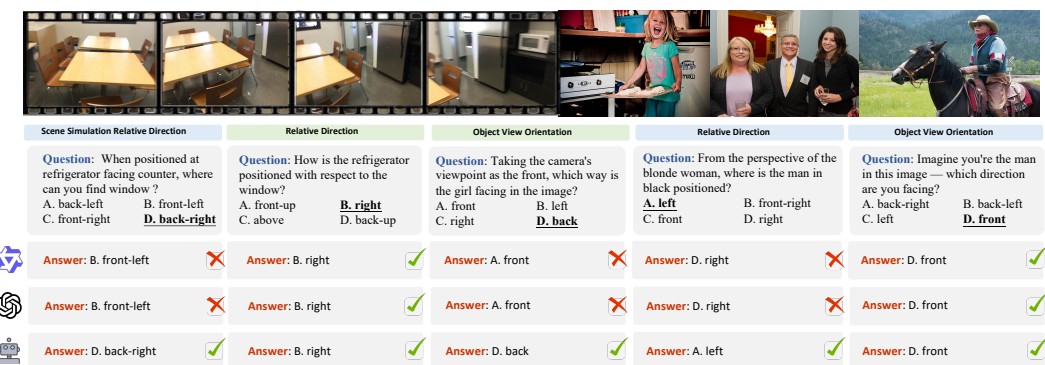

Figure 8: ViewSpatial-Bench Examples (Part2).

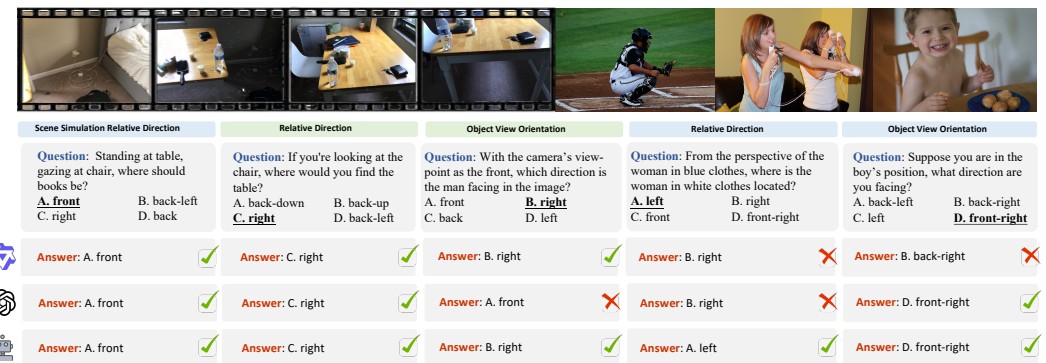

Figure 9: ViewSpatial-Bench Examples (Part3).

