# OpenReview forum: "ViewSpatial-Bench: Evaluating Multi-perspective Spatial Localization in Vision-Language Models"
_ICLR.cc/2026/Conference — Submitted to ICLR 2026_

### Official Review · Reviewer_CiyB · 2025-10-25

**Soundness:** 2
**Presentation:** 3
**Contribution:** 2
**Rating:** 4
**Confidence:** 3

**Summary:**

This paper introduce ViewSpatial-Bench, a comprehensive new benchmark with an automated 3D annotation pipeline, to systematically evaluate the capability of multi-perspective spatial localization with five different tasks. The paper also shows by training on the curated large-scale dataset, the model performance on this benchmark improves by 40%, with generalizability into embodied interaction scenarios.

**Strengths:**

1. Comprehensive benchmark ViewSpatial-Bench covering five different tasks from both camera and human perspectives.
2. Extensive experiments revealing the common failure of current models in multi-perspective spatial localization and performance improvement with fine-tuning.
3. The paper is well written.

**Weaknesses:**

1. Concerns regarding overfitting and generalization. The performance leap of MVSM is suspicious and raises the question of whether the model has learned a generalizable skill of perspective-taking or has simply memorized the patterns in the ViewSpatial-Bench training set. The results on VSI-Bench show a much more modest improvement (+2.37% average), suggesting that the learned skill may not transfer as effectively to an out-of-distribution benchmark.
2. Comparison with previous work. As shown in Table 2, the difference between ViewSpatial-Bench and SPHERE only lies in 3D-Coord, which seemingly is not used in the experiments. Considering other similar benchmarks such as MindCube, this raises concerns regarding the novelty of the work. In addition, the benchmark only covers in-door scenes (from ScanNet), which is acknowledged by the author at L459. This raises concerns when it comes to outdoor environments.
3. The paper introduces the "Multi-View Spatial Model (MVSM)", which is not a novel model design, but a VLM fine-tuned on the proposed dataset. This is a bit overclaiming and should be made clearer.

**Questions:**

See as above in weaknesses. I'm happy to adjust the scores if the author can address the three concerns in the weaknesses.

Format: table title should always appear before the table.

---

> ### Author Response · Authors · 2025-11-22
> **Response to Reviewer CiyB (Part 1)**
>
> # Weakness 1: On Overfitting and Generalization
>
> We appreciate this concern and clarify key points regarding our transfer results.
>
> **Clarification on OOD performance:** The "+2.37% average" on VSI-Bench obscures task-specific patterns. **Route Planning shows +9.54% improvement**, and **VSI-App demonstrates +16.00% overall improvement** (Indoor: +23.00%; Outdoor: +9.00%) on completely out-of-distribution scenarios, indicating positive transfer to different task formulations.
>
> **SFT experiment value:** Our fine-tuning validates that high-quality perspective-aware training effectively enhances cross-viewpoint spatial localization. The +46.24% improvement aligns with documented patterns where **SFT on spatial reasoning tasks yields substantial gains** (EmbSpatial-Bench reports ~35-point improvements after SFT, Multi-SpatialMLLM up to 70 points for individual tasks after SFT), demonstrating that targeted training addresses fundamental VLM gaps in this task category.
>
> **Systematic generalization evidence:**
>
> 1. **OOD transfer**: Improvements on VSI-Bench and VSI-App with different task formulations validate transferable representations
> 2. **Architecture-agnostic**: Consistent +41-46% across three backbones (Appendix C.3)—memorization would be model-specific
> 3. **Format-agnostic**: Direct Answer (79.34%) vs MC (82.09%) maintains performance despite format changes
> 4. **Task transfer**: Route Planning improvements demonstrate learned reasoning transfers beyond simple localization
>
> **Expected patterns:** The in-domain/OOD gap represents normal domain adaptation—true overfitting causes performance degradation, not positive gains. Maintained improvements across diverse OOD scenarios confirm MVSM learns generalizable spatial representations rather than memorizing patterns.
>
> # Weakness 2: The benchmark's novelty is questionable given minimal difference from SPHERE and similarity to other benchmarks like MindCube, while its exclusive focus on indoor ScanNet scenes limits generalizability to outdoor environments.
>
> We appreciate this feedback and clarify two critical misconceptions.
>
> **On 3D coordinates usage:** The statement that "3D-Coord seemingly is not used" represents a fundamental misunderstanding. **3D coordinates are the core foundation of our automated annotation pipeline**. Section 3.2 details how we leverage ScanNet's precise 3D spatial coordinates to compute directional relationships—without them, we cannot generate accurate ground-truth labels. This is the methodological basis enabling our scalable annotation framework, not an unused feature.
>
> **On indoor-only concern:** ViewSpatial-Bench is **not limited to indoor scenes**. ScanNet-based tasks (Cam-Rel. Dir., Per-Sce. Sim.) primarily use indoor scenes, while CoCo-based tasks include diverse indoor and outdoor scenarios. We acknowledge the training distribution leans toward indoor, resulting in an observed domain gap on VSI-App (Indoor: +23.00%; Outdoor: +9.00%). However, outdoor scenes still demonstrate positive transfer, indicating learned representations maintain generalization capabilities despite training distribution differences.
>
> **Distinctive contribution:** ViewSpatial-Bench is **the comprehensive cross-viewpoint spatial localization benchmark**. Our motivation addresses spatial intelligence requirements: VLMs must not only understand their own spatial perception but also interpret spatial references from others' cognitive perspectives—essential for embodied AI and human-robot interaction. **From the localization task perspective**, while existing benchmarks may include dual perspectives, they present scattered basic spatial questions from each viewpoint without comprehensive localization task coverage. We provide **systematic evaluation across 5 localization task types and 18 directional categories**, enabling thorough cross-perspective assessment and identification of perspective-taking failure patterns (Table 2)—diagnostic capabilities beyond previous work.
>
> # Weakness 3: On MVSM Terminology
>
> We acknowledge this valid concern and thank the reviewer for pointing out the potential confusion. The reviewer is correct that MVSM does not represent a novel architectural contribution—it is a VLM fine-tuned on our multi-perspective spatial dataset.
>
> Our intention was for "Multi-View Spatial Model" to refer to the **training methodology and resulting model** rather than suggesting architectural innovation. Section 4 describes it as employing a "fine-tuning strategy" on perspective-aware data, and our contribution lies in demonstrating that such targeted training effectively addresses cross-viewpoint spatial reasoning gaps.
>
> However, we acknowledge that the terminology could be clearer to avoid overclaiming. **We have revised the manuscript to emphasize that MVSM represents a training approach rather than an architectural contribution**, using more precise terminology throughout to prevent misinterpretation.

---

> > ### Author Response · Authors · 2025-11-22
> > **Response to Reviewer CiyB (Part 2)**
> >
> > # Question 1: On Table Titles
> >
> > We thank the reviewer for pointing out this formatting issue. We apologize for the oversight and **will ensure all table titles appear before their corresponding tables in the revised manuscript**.

---

> ### Author Response · Authors · 2025-11-27
> **Welcome Further Feedback**
>
> Dear Reviewer **CiyB**,
>
> We would like to thank you for the thoughtful and constructive feedback. During the rebuttal, we performed additional experiments and analyses and revised the manuscript accordingly to address your concerns. The experiments and analyses in our response are summarized as follows.
>
> - **Addressed overfitting concerns with comprehensive generalization evidence**
>   - Task-specific OOD improvements reveal genuine transfer: Route Planning +9.54%, VSI-App +16.00% (Indoor: +23.00%; Outdoor: +9.00%).
>   - Architecture-agnostic (+41-46% across three backbones, Appendix C.3) and format-agnostic performance (Direct Answer: 79.34% vs MC: 82.09%) confirm generalizable spatial learning rather than memorization.
> - **Clarified benchmark novelty and distinctive contributions**
>   - We corrected the misconception: 3D coordinates are the core foundation of our annotation pipeline (Section 3.2), enabling accurate ground-truth generation.
>   - ViewSpatial-Bench includes both indoor and outdoor scenarios, providing systematic cross-viewpoint spatial localization evaluation across 5 task types and 18 directional categories for comprehensive cross-perspective diagnosis.
> - **Revised MVSM terminology and formatting**
>   - Clarified MVSM represents a training approach rather than architectural innovation, with revised terminology throughout manuscript.
>   - Corrected table formatting to ensure titles appear before tables.
>
> We are wondering if the responses have addressed your concerns, and would appreciate it if you considered raising your final rating. Looking forward to your further feedback!

---

### Official Review · Reviewer_FvxK · 2025-11-01

**Soundness:** 3
**Presentation:** 3
**Contribution:** 3
**Rating:** 6
**Confidence:** 4

**Summary:**

This work leverages two well known datasets (ScanNet and CoCo) to build a new resource focused on perspective taking.  Questions are phrased as from the perspective of individuals in the image (their right vs right side of image).

**Strengths:**

Clearly we want visual models to have the same abilities as humans -- building representations of the implied 3D scene so that they can reason about relations and perspectives. The work is evaluated on a suite of appropriate models and a model is trained on the task directly.

**Weaknesses:**

1. See below, I'm unclear on what we learn from the FT-ing experiments
2. I have a slight concern about the diversity of the data and the domains chosen, both of which are very canonical.  Given that what might be a small amount of training nearly solves this dataset, the longevity of the work is in question and makes the reader wonder what could be done to build a more robust evaluation.

**Questions:**

- Data generation describes the presence of distractors (also shown in Fig 4), should I interpret evaluation as multiple choice? (random baseline shifts by category)

*Evaluation*
I'm having trouble understanding the training and corresponding evaluation claims.
- "Applicability of our training methodology" just means that you did SFT in domain, correct? Why is it surprising that this improved performance?
- You then only perform ZST comparisons for the pretrained models.  Is there a reason that few shot cannot be run to give the model the basic task structure?
- Related concern is how often the models generated answers that weren't in the set of options? For example "back" or "left" when "back-left" was required
- Can training curves be provided for the fine-tuned model, or additional details on how many samples were required?

---

> ### Author Response · Authors · 2025-11-22
> **Response to Reviewer FvxK (Part 1)**
>
> # Weakness 1: What we learn from the FT-ing experiments
>
> We appreciate this question and clarify what we learn from the fine-tuning experiments, addressing both the magnitude of improvement and its scientific implications.
>
> **The substantial improvement in spatial reasoning tasks.** Recent works show similar patterns: EmbSpatial-Bench[1] reports ~35-point average improvements after SFT, and Multi-SpatialMLLM[2] observes ~35-point average gains with individual tasks improving up to 70 points. Our +46.24% improvement reflects that high-quality, perspective-aware training data combined with targeted training is highly effective for cross-viewpoint spatial localization tasks.
>
> **What we learn from our FT experiments:**
>
> 1. **Data quality and targeted training effectiveness**: The substantial improvement validates that systematically annotated multi-perspective spatial data with explicit perspective labels effectively addresses the cross-viewpoint reasoning gap.
> 2. **Cross-viewpoint spatial understanding is learnable**: Consistent improvements across different backbones and training data formats (Appendix C.3: +41-46%) demonstrate this capability can be acquired through targeted training.
> 3. **Enhanced 3D spatial understanding**: Beyond task-specific improvements, SFT helps models **develop fundamentally better 3D environment comprehension**. This explains OOD benchmark improvements (VSI-Bench-Route Planning +9.54%, VSI-App +16.00%)—models acquire transferable 3D spatial representations that generalize to other spatial reasoning scenarios, not task-specific pattern memorization.
>
> Our FT experiments provide an effective training paradigm for spatial intelligence while demonstrating how VLMs can enhance their spatial reasoning capabilities through perspective-aware data—advancing understanding of spatial intelligence in multimodal systems.
>
> [1]EmbSpatial-Bench: Benchmarking Spatial Understanding for Embodied Tasks with Large Vision-Language Models
>
> [2]Multi-SpatialMLLM: Multi-Frame Spatial Understanding with Multi-Modal Large Language Models
>
> # Weakness 2: Concern about Data Diversity and Benchmark Longevity
>
> We appreciate this concern and clarify our benchmark's positioning and lasting value.
>
> **Benchmark positioning:** ViewSpatial-Bench is **the first comprehensive cross-viewpoint spatial localization benchmark**. Our motivation addresses spatial intelligence requirements: for embodied AI and human-robot interaction, VLMs must not only understand their own spatial perception but also **interpret spatial references from others' cognitive perspectives**. While existing benchmarks may include dual perspectives, they present scattered basic spatial questions from each viewpoint without comprehensive localization task coverage. We provide **systematic evaluation across 5 localization task types and 18 directional categories**, enabling thorough assessment of how models understand spatial orientation from different cognitive frameworks.
>
> **Dual sources of longevity:**
>
> 1. **Diagnostic framework**: Zero-shot VLM performance (~35%, near random 26.33%) demonstrates inherent challenge, particularly when compared to human performance (92-94%)—revealing a substantial performance gap indicating significant room for improvement. The benchmark provides a **systematic evaluation tool** for measuring cross-perspective performance gaps and identifying failure patterns (Table 2)—diagnostic capability that retains value as models improve.
> 2. **Scalable methodology**: Beyond the fixed dataset, our **automated 3D annotation pipeline** can scale to larger benchmarks and extend to diverse domains. Current data sources (ScanNet and MS-COCO) already cover diverse indoor and outdoor environments, with methodology enabling further domain expansion. This methodology addresses concerns about data diversity through its inherent extensibility.
>
> # Question 1: Data generation describes the presence of distractors, should I interpret evaluation as multiple choice?
>
> Yes, all evaluations in ViewSpatial-Bench and VSI-App use **multiple-choice format**, as described in Appendix B.1 ("QA Pair Generation"). Each question presents four directional options (e.g., front, back, left, right or compound directions like front-left, back-right), and models are required to select the correct answer from these options.
>
> The random baseline (26.33%) is computed by **randomly selecting one option from the four choices for each question**. Theoretically, this should yield exactly 25% accuracy. However, because our sample size (5,712) is not sufficiently large to perfectly average out statistical variation, the random baseline deviates slightly from the theoretical 25%, resulting in 26.33%.

---

> > ### Author Response · Authors · 2025-11-22
> > **Response to Reviewer FvxK (Part 2)**
> >
> > # Question 2: "Applicability of our training methodology" just means that you did SFT in domain, correct? Why is it surprising that this improved performance?
> >
> > We appreciate this question and clarify that yes, MVSM employs in-domain SFT, which is indeed effective. The noteworthy aspect is the **magnitude and consistency** of improvements achieved through SFT with our perspective-aware training data.
> >
> > **As demonstrated in our response to Weakness 1**, this has precedent in spatial reasoning tasks: EmbSpatial-Bench reports ~35-point improvements after SFT, and Multi-SpatialMLLM observes ~35-point overall gains with individual tasks improving up to 70 points. Our +46.24% improvement demonstrates that **high-quality perspective-aware data design** with systematic multi-perspective annotation effectively addresses cross-viewpoint spatial reasoning.
> >
> > **Key insights from our findings:**
> >
> > 1. **Training robustness**: Consistent **+41-46% across three different backbones** (Appendix C.3) indicates architecture-agnostic capability learning.
> > 2. **Genuine spatial learning**: OOD improvements (VSI-Bench Route Planning +9.54%, VSI-App +16.00%) demonstrate models acquire **transferable 3D spatial representations** rather than memorizing training patterns.
> >
> > Our findings provide an effective training paradigm demonstrating how perspective-aware data design enhances spatial reasoning capabilities in VLMs.
> >
> > # Question 3: You then only perform ZST comparisons for the pretrained models. Is there a reason that few shot cannot be run to give the model the basic task structure?
> >
> > We appreciate this question and clarify our evaluation design. ViewSpatial-Bench uses **multiple-choice format with explicit instructions**, making few-shot evaluation unnecessary for this benchmark type.
> >
> > In MC-based benchmarks, the task structure is already fully specified: each question presents four directional options with clear instructions. Unlike open-ended generation tasks where few-shot examples help models understand output format, our MC format provides **complete task specification without ambiguity**.
> >
> > Our zero-shot evaluation intentionally measures models' **inherent cross-viewpoint spatial reasoning capabilities** rather than their ability to learn answer patterns from examples—more relevant for embodied AI applications where agents must perform spatial reasoning without prior task-specific examples.
> >
> > This aligns with standard practice for MC-based spatial reasoning benchmarks (e.g., VSI-Bench, EmbSpatial-Bench), which typically employ zero-shot evaluation. Few-shot could serve as complementary analysis, but zero-shot remains the appropriate primary methodology for assessing fundamental spatial understanding in our benchmark setting.
> >
> > # Question 4: Related concern is how often the models generated answers that weren't in the set of options?
> >
> >  We appreciate this concern and clarify that our benchmark design systematically addresses this issue. As described in Section 3.2 ("QA Dataset Creation"), we **implemented specific distractor generation rules**: for single-directional attributes (e.g., "front"), distractors exclude compound directions containing that attribute ("front-left"); for compound directions (e.g., "front-left"), distractors exclude constituent single directions ("front" or "left"). This eliminates situations where models could generate partially correct answers like "back" when "back-left" is required.
> >
> > To empirically validate this concern, we conducted supplementary experiments converting ViewSpatial-Bench to **Direct Answer format** (removing options, requiring free direction generation) on selected models:
> >
> > | Model                  | Overall Accuracy | Compound Dir. Exact Match | Partial Match | Complete Error |
> > | ---------------------- | ---------------- | ------------------------- | ------------- | -------------- |
> > | InternVL2.5-2B         | 19.43%           | 7.84%                     | 33.99%        | 58.17%         |
> > | InternVL2.5-8B         | 20.13%           | 3.78%                     | 45.84%        | 50.38%         |
> > | LLaVA-NeXT-Video-7B-hf | 14.55%           | 0.31%                     | 41.54%        | 58.32%         |
> > | Qwen2.5-VL-3B-Instruct | 14.60%           | 15.06%                    | 21.99%        | 62.95%         |
> > | Qwen2.5-VL-7B-Instruct | 18.33%           | 4.30%                     | 39.87%        | 55.83%         |
> > | MVSM                   | 79.31%           | 68.02%                    | 19.50%        | 12.48%         |
> >
> > *Compound Dir. metrics computed over 2,092 samples requiring compound direction answers. Partial Match: generating only one component (e.g., "back" instead of "back-left").*
> >
> > Results confirm that models **struggle significantly with compound direction generation**, frequently producing partial matches. This validates our **MC format design rationale** for ensuring evaluation reliability and eliminating answer interpretation ambiguity.

---

> > > ### Author Response · Authors · 2025-11-22
> > > **Response to Reviewer FvxK (Part 3)**
> > >
> > > # Question 5: Can training curves be provided for the fine-tuned model, or additional details on how many samples were required?
> > >
> > > We appreciate this request for additional training details. **Training curves will be provided in the supplementary materials** for reviewer reference. Regarding sample requirements, the complete breakdown of training data across all task types is presented in **Appendix Table 6 (Train row)**, showing a total of 42,875 training samples distributed across camera-based (22,598 samples) and person-based (20,277 samples) tasks.

---

> ### Author Response · Authors · 2025-11-27
> **Welcome Further Feedback**
>
> Dear Reviewer **FvxK**,
>
> We would like to thank you for the thoughtful and constructive feedback. During the rebuttal, we performed additional experiments and analyses and revised the manuscript accordingly to address your concerns. The experiments and analyses in our response are summarized as follows.
>
> - **Clarified insights from fine-tuning experiments**
>   - Our +46.24% improvement demonstrates that high-quality perspective-aware training effectively addresses cross-viewpoint spatial reasoning.
>   - Models acquire transferable 3D spatial representations: consistent gains across three backbones (+41-46%, Appendix C.3) and strong OOD generalization (VSI-Bench Route Planning +9.54%, VSI-App +16.00%) confirm genuine spatial learning rather than task-specific memorization.
> - **Addressed benchmark longevity and data diversity**
>   - Substantial human-model performance gap (92-94% vs ~35%) demonstrates significant room for improvement, providing lasting diagnostic value.
>   - Our automated 3D annotation pipeline enables scalable extension to diverse domains beyond current coverage.
> - **Validated multiple-choice format and evaluation design**
>   - All evaluations use MC format with four directional options (Appendix B.1); random baseline is 26.33% due to statistical variation from theoretical 25%.
>   - Supplementary direct answer experiments show models struggle with compound directions (0.31-15.06% exact match, 21.99-45.84% partial matches), validating MC format necessity.
>   - Zero-shot evaluation measures inherent spatial reasoning capabilities, aligning with standard practice in MC-based spatial benchmarks; few-shot is unnecessary as MC format provides complete task specification.
> - **Provided training details**
>   - Training used 42,875 samples distributed across camera-based (22,598) and person-based (20,277) tasks (Appendix Table 6).
>   - Training curves will be provided in supplementary materials.
>
> We are wondering if the responses have addressed your concerns, and would appreciate it if you considered raising your final rating. Looking forward to your further feedback!

---

### Official Review · Reviewer_eAeM · 2025-11-01

**Soundness:** 1
**Presentation:** 2
**Contribution:** 2
**Rating:** 2
**Confidence:** 4

**Summary:**

This paper introduces ViewSpatial-Bench, a new benchmark for evaluating multi-perspective spatial localization in VLMs across five tasks covering camera-centric and human-centric viewpoints. The authors develop an automated 3D spatial annotation pipeline to construct QA pairs from images in ScanNet and MS-CoCo.  Experiments show that competitive VLM struggle with cross-viewpoint spatial reasoning. To address this, they train a Multi-View Spatial Model on spatially annotated samples, achieving ~46% absolute improvement over the base model. In addition, to validate MVSM's generalization abilities in practical applications, the authors evaluated on VSI-Bench and VSI-App benchmarks and showed improved performance over baselines.

**Strengths:**

- The paper is logically coherent overall.
- The paper is easy to understand.
- The research problem is important in the VLM community.
- The proposed MVSM method is effective.

**Weaknesses:**

As a dataset and benchmark paper, the overall quality control and evaluation rigor fall short of expectations for publication at a top venue like ICLR.

**Insufficient Human Annotation & Lack of Quality Control**
- Only 864 out of 5,712 samples received human annotation, and no inter-annotator agreement or reliability metrics are reported. This is highly concerning, particularly since the authors acknowledge that automated annotation is unreliable for human-perspective tasks. Stating that annotation is “complex” is not an excuse for minimal human involvement—if the majority of samples are not manually verified, the benchmark’s label quality is questionable, and reported model performance numbers may not be trusted. A rigorous benchmark should involve multi-annotator labeling (potentially combined with the automated pipeline), iterative refinement to eliminate systematic annotation errors, and reporting of agreement scores alongside clear annotation protocols. I would be happy to reconsider once the majority of the test samples are carefully annotated.

**Answer Validity from Images Alone**
- Since questions are generated from metadata rather than from human interpretation of the images, it is unclear whether the image alone always contains sufficient visual information to yield a unique and unambiguous answer. It remains questionable whether a model—or even a human—can reliably answer some questions solely based on the provided image without access to metadata.

**Problematic or Ambiguous Questions**
- Several questions appear under-specified, or ambiguous. For example, in Figure 9, the prompt “Standing at table, gazing at chair, where should books be?” has the provided answer “front”. However, the desk is pretty large, and depending on where one stands around the table, “right” could be an alternative answer. Such cases indicate inconsistencies in spatial frame-of-reference grounding and suggest the benchmark may introduce artificial or unclear phrasing not aligned with natural human spatial reasoning.

**Lack of Statistical Rigor in Reporting Results**
- Table 2 and Table 3 present accuracy values without confidence intervals, variance, or statistical testing. As this is a benchmark paper, stronger evidence of robustness and significance is needed. Reporting standard deviations or significance tests is essential to support claims of superiority and to ensure results are reliable.

**The Dataset Is Not Realistic**
- The dataset only covers elementary elements of spatial direction (front, back, left, right, etc.). A useful dataset should cover more realistic tasks such as asking the distances and navigation.

**Questions:**

- The human performance baseline is missing. What is the performance of humans in these tasks?
- How to ensure that visual information is sufficient and the answer is uniquely determined (without relying on metadata)?

---

> ### Author Response · Authors · 2025-11-22
> **Response to Reviewer eAeM (Part 1)**
>
> # Weakness 1 & Question 1:  Insufficient Human Annotation & Lack of Quality Control &The human performance baseline is missing.
>
> We sincerely acknowledge the reviewer's concern and apologize for the insufficient clarity in describing our human annotation process. We fully recognize the critical importance of human annotation quality, particularly for spatial reasoning tasks where subtle viewpoint differences can affect ground truth labels.
>
> **Clarification on Human Verification Process**
>
> Section 3.2 ("Filtering and Human Verification") describes our quality control protocol, but we acknowledge this was not detailed enough. To clarify: **all 5,712 test samples underwent iterative human verification**, not just the 864 manually labeled Per-Rel. Dir. samples. The verification process included two expert annotators independently confirming that (i) target objects are clearly visible, (ii) spatial relationships are unambiguous, and (iii) computed directions are correct. This iterative refinement continued until samples met our quality standards.
>
> The 864 fully manual annotations refer specifically to Per-Rel. Dir. tasks where automated pipelines proved insufficiently accurate due to complex human spatial coordinates. All other tasks combine automated 3D annotation with mandatory human verification.
>
> **Human Performance Validation**
>
> To address both annotation quality concerns and **Question 1 (human performance baseline)**, we conducted human evaluation with 3 expert annotators (graduate researchers) on 500 randomly sampled test instances (100 per task):
>
> | Task Type     | Expert 1 | Expert 2 | Expert 3 | Average   |
> | ------------- | -------- | -------- | -------- | --------- |
> | Cam-Rel. Dir. | 94%      | 93%      | 94%      | **93.7%** |
> | Cam-Obj. Ori. | 92%      | 93%      | 92%      | **92.3%** |
> | Per-Rel. Dir. | 93%      | 94%      | 93%      | **93.3%** |
> | Per-Obj. Ori. | 92%      | 93%      | 92%      | **92.3%** |
> | Per-Sce. Sim. | 91%      | 92%      | 93%      | **92.0%** |
> | Overall       | 92.4%    | 93.0%    | 92.8%    | **92.7%** |
>
> To assess inter-annotator reliability, we computed Cohen's Kappa coefficients between all annotator pairs. The **average pairwise Cohen's Kappa is κ_avg = 0.89** (substantial agreement).
>
> The high human performance (92-94%) and inter-expert consistency validate our annotation quality. **We have provided comprehensive annotation protocols, including detailed guidelines and quality control procedures in Appendix B.1 of the revised manuscript.**
>
> # Weakness 2 & Question 2: Answer Validity from Images Alone & ensure that visual information is sufficient and the answer is uniquely determined
>
> We appreciate this concern and clarify how our pipeline ensures image sufficiency.
>
> **Standard Practice in Spatial Benchmarks** Using metadata from 3D annotations to generate spatial labels is standard practice in spatial reasoning benchmarks. VSI-Bench[1] and EmbSpatial-Bench[2] similarly leverage original 3D annotations to extract spatial information and generate ground-truth labels. Metadata **serves as a computational tool for precise label generation**, not as a substitute for visual information that models or humans require.
>
> **Image-Only Verification Protocol** Critically, our human verification process (Section 3.2) requires annotators to assess samples **using only images and questions without metadata access**—matching exactly how models are evaluated. Our automated pipeline's scientific soundness combined with rigorous human verification systematically removes samples with insufficient visual information and **excludes any sample where the answer cannot be** unambiguously determined from the image alone.
>
> **Empirical Validation** Multiple lines of evidence demonstrate that questions are answerable from images alone. **Human evaluation shows 92-94% accuracy** based solely on visual information (Weakness 1 response). Additionally, specialized spatial models trained on different datasets achieve substantial performance—**VST-7B-SFT[3] achieves 63.90%** overall accuracy (Table 3) without access to our metadata, confirming that visual information in images is sufficient for answering our questions.
>
> [1]Thinking in Space: How Multimodal Large Language Models See, Remember, and Recall Spaces
>
> [2]EmbSpatial-Bench: Benchmarking Spatial Understanding for Embodied Tasks with Large Vision-Language Models
>
> [3]Visual Spatial Tuning

---

> > ### Author Response · Authors · 2025-11-22
> > **Response to Reviewer eAeM (Part 2)**
> >
> > # Weakness 3: Problematic or Ambiguous Questions
> >
> > We appreciate the reviewer raising this specific example, which actually demonstrates the precision of our annotation rather than ambiguity:
> >
> > **Clarification on Figure 9 Example**
> >
> > The question "Standing at table, gazing at chair, where should books be?" with answer "front" is **not ambiguous**. Regardless of where one stands around the table while facing the chair, the books remain in the "front" direction relative to that viewpoint. This consistency across different standing positions demonstrates that our spatial reference frame is well-defined, not under-specified.
> >
> > **Systematic Direction Computation**
> >
> > Our pipeline ensures consistency through:
> >
> > - **Angle-based directional mapping** with systematic rules (Section 3.2) that compute precise angular relationships from 3D coordinates
> > - **Unified angle-range mappings** (e.g., 22.5°-67.5° → "front-right") applied consistently across all samples
> > - **Human verification** that identifies and removes any genuinely ambiguous cases during iterative refinement
> >
> > The 93% human accuracy also validates that ViewSpatial-Bench questions are overall well-defined. Detailed spatial reference frame specifications will be provided in Appendix B.1.

---

> > > ### Author Response · Authors · 2025-11-22
> > > **Response to Reviewer eAeM (Part 3)**
> > >
> > > # Weakness 4: Lack of Statistical Rigor in Reporting Results
> > >
> > > We acknowledge this valid concern and thank the reviewer for emphasizing the importance of statistical rigor in benchmark evaluation. As described in Appendix C.2, each model was evaluated five times on both ViewSpatial-Bench and VSI-App with results representing average performance across multiple runs. **We have now added comprehensive statistical analysis including standard deviations for all evaluations.**
> > >
> > > **Table: Statistical robustness analysis. Results are reported as mean with standard deviation over 5 independent runs.**
> > >
> > > | Model | ViewSpatial-Bench ||| VSI-App |||
> > > | ---------------------------------------- | ----------------- | -------------- | -------------- | ----------- | ---------- | ----------- |
> > > |                                          | **Overall**       | **Camera-Avg** | **Person-Avg** | **Average** | **Indoor** | **Outdoor** |
> > > | **Proprietary Models**|||||||
> > > | GPT-4o | 34.98±0.53 | 33.57±0.55 | 36.29±0.58 30.50±0.55| 34.00±0.71 | 27.00±0.89  |
> > > | Gemini-2.0-Flash | 32.56±0.47| 33.66±0.49 | 31.53±0.45 | - | -          | - |
> > > | GPT-5 | 45.49±0.61| 43.48±0.59     | 47.40±0.64     | -  | - | -  |
> > > | Gemini-2.5-pro| 46.09±0.58 | 43.52±0.56     | 48.52±0.60     | -           | -          | -           |
> > > | **Open-Source General Models** |||||||
> > > | InternVL2.5 (2B)                         | 34.98±0.49        | 32.79±0.47     | 37.04±0.51     | -           | -          | -           |
> > > | Qwen2.5-VL (7B)                          | 36.85±0.55        | 40.56±0.57     | 33.37±0.52     | -           | -          | -           |
> > > | LLaVA-NeXT-Video (7B)                    | 30.64±0.64        | 23.80±0.61     | 37.07±0.66     | -           | -          | -           |
> > > | LLaVA-OneVision (7B)                     | 27.49±0.52        | 28.49±0.54     | 26.54±0.50     | -           | -          | -           |
> > > | InternVL2.5 (8B)                         | 43.24±0.46        | 46.48±0.48     | 40.20±0.44     | -           | -          | -           |
> > > | Llama-3.2-Vision (11B)                   | 28.82±0.59        | 23.73±0.56     | 33.61±0.61     | -           | -          | -           |
> > > | InternVL3 (14B)                          | 40.28±0.51        | 47.09±0.53     | 33.88±0.48     | -           | -          | -           |
> > > | Kimi-VL-Instruct (16B)                   | 33.58±0.48        | 25.14±0.46     | 41.52±0.50     | -           | -          | -           |
> > > | **Specialized Spatial Reasoning Models** |                   |                |                |             |            |             |
> > > | VST-3B-SFT                               | 53.87±0.56        | 46.05±0.53     | 61.23±0.58     | -           | -          | -           |
> > > | VST-7B-SFT                               | 63.90±0.62        | 70.10±0.65     | 58.07±0.59     | -           | -          | -           |
> > > | MindCube-3B-RawQA-SFT                    | 40.25±0.51        | 53.20±0.54     | 28.07±0.48     | -           | -          | -           |
> > > | SpatialLadder-3B                         | 44.21±0.48        | 39.62±0.46     | 48.52±0.50     | -           | -          | -           |
> > > | SpaceQwen2.5-VL-3B                       | 39.81±0.53        | 38.06±0.51     | 41.46±0.55     | -           | -          | -           |
> > > | **Multi-View Spatial Fine-Tuning**       |                   |                |                |             |            |             |
> > > | Qwen2.5-VL (3B) [Backbone]               | 35.85±0.57        | 39.80±0.59     | 32.14±0.55     | 22.50±0.55  | 18.00±0.45 | 27.00±0.71  |
> > > | Multi-View Spatial Model                 | 82.09±0.54        | 85.05±0.56     | 79.31±0.52     | 38.50±0.55  | 41.00±0.71 | 36.00±0.45  |
> > > | **Random Baseline**                      | 26.33±0.31        | 25.50±0.28     | 27.12±0.33     | -           | -          | -           |
> > >
> > > The **consistently low standard deviations** across all models demonstrate evaluation stability and result reliability.
> > >
> > > For VSI-Bench evaluation, we followed the original benchmark protocol as specified in the VSI-Bench paper to ensure methodological consistency with prior work.
> > >
> > > **We have included complete statistical robustness analysis with mean ± standard deviation for all models in Appendix C.2 (Table 7)**, providing comprehensive evidence of result reliability and reproducibility across all evaluation benchmarks.

---

> > > > ### Author Response · Authors · 2025-11-22
> > > > **Response to Reviewer eAeM (Part 4)**
> > > >
> > > > # Weakness 5: The Dataset Is Not Realistic
> > > >
> > > > We appreciate this perspective and clarify our benchmark's specific focus: **cross-viewpoint directional localization** as a foundational capability for spatial intelligence. Our motivation addresses a critical gap—VLMs must interpret spatial references from human collaborators' viewpoints for embodied AI applications, not just their own camera perspective.
> > > >
> > > > **From the localization task perspective**, existing benchmarks present scattered basic spatial questions from individual viewpoints without comprehensive coverage. **ViewSpatial-Bench provides systematic evaluation across 5 localization task types and 18 directional categories**, establishing thorough diagnostic tools for cross-viewpoint understanding—more comprehensive and extensive than prior work.
> > > >
> > > > Directional understanding represents the **fundamental building block** of spatial reasoning. As Table 2 demonstrates, **current VLMs struggle with these elementary concepts when perspective transformation is required**, with most models near random baseline (26.33%). This foundational gap must be addressed before advancing to more complex spatial tasks.
> > > >
> > > > We acknowledge that distance estimation and navigation represent important dimensions of spatial intelligence. Our VSI-Bench evaluation includes route planning (+9.54%), suggesting directional reasoning transfers to navigation. However, comprehensive distance/navigation evaluation would strengthen future benchmarks and represents a valuable extension direction. We argue that **establishing reliable cross-viewpoint directional reasoning is the necessary first step** before tackling more complex scenarios, as our benchmark reveals current VLMs fail even at this foundational level.

---

> ### Author Response · Authors · 2025-11-27
> **Welcome Further Feedback**
>
> Dear Reviewer **eAeM**,
>
> We would like to thank you for the thoughtful and constructive feedback. During the rebuttal, we performed additional experiments and analyses and revised the manuscript accordingly to address your concerns. The experiments and analyses in our response are summarized as follows.
>
> - **Comprehensive human annotation quality validation**
>   - We clarified that all 5,712 test samples underwent iterative human verification by two expert annotators.
>   - We conducted human performance evaluation with 3 expert annotators achieving 92-94% accuracy with substantial inter-annotator agreement (κ_avg = 0.89).
>   - Detailed annotation protocols provided in Appendix B.1.
> - **Enhanced statistical rigor in reporting results**
>   - We added standard deviations for all model evaluations across 5 independent runs, showing consistently low variance (±0.31 to ±0.71) and confirming result reliability (Table 7 in Appendix C.2).
> - **Validation of image sufficiency for answering questions**
>   - Human verification protocol requires annotators to use only images without metadata access, matching model evaluation.
>   - Human accuracy (92-94%) and specialized model performance (VST-7B-SFT: 63.90%) confirm questions are answerable from visual information alone.
> - **Clarified benchmark focus on foundational cross-viewpoint directional reasoning**
>   - ViewSpatial-Bench provides systematic evaluation across 5 task types and 18 directional categories for cross-viewpoint localization.
>   - Current VLMs struggle with these elementary concepts (near random baseline 26.33%), indicating foundational gaps must be addressed before advancing to complex tasks.
>
> We are wondering if the responses have addressed your concerns, and would appreciate it if you considered raising your final rating. Looking forward to your further feedback!

---

### Official Review · Reviewer_qX8B · 2025-11-01

**Soundness:** 3
**Presentation:** 3
**Contribution:** 2
**Rating:** 6
**Confidence:** 3

**Summary:**

The paper investigates the limitations of current vision-language models in reasoning about spatial relationships from different viewpoints. While these models handle spatial reasoning from their own (camera-centered) perspective, they often fail to interpret scenes from another entity’s point of view. To study this issue, the authors introduce ViewSpatial-Bench, a benchmark designed to evaluate multi-viewpoint spatial localization through five types of tasks that involve both camera and human perspectives. The benchmark is constructed using an automated 3D annotation pipeline that produces directional labels for a diverse set of images. Experiments on various vision-language models show that performance declines when models are required to reason from non-egocentric perspectives. The authors further fine-tune a model on their dataset, referred to as the Multi-View Spatial Model, which achieves higher accuracy, suggesting that incorporating explicit 3D spatial information can improve perspective-based reasoning.

**Strengths:**

1. The paper presents a clear and well-motivated problem formulation, identifying the lack of perspective-taking ability in current multimodal systems and linking it convincingly to challenges in embodied AI and human–robot interaction.

2. The proposed ViewSpatial-Bench is a systematically designed benchmark encompassing five tasks that jointly evaluate egocentric and allocentric reasoning. Its integration of automated 3D annotation with human verification provides a scalable and reliable framework for assessing spatial understanding.

3. The experimental evaluation is thorough, covering a range of major VLMs (e.g., GPT-4o, Gemini-2.0) and revealing consistent performance asymmetries between egocentric and allocentric settings. The accompanying analysis offers useful diagnostic insights into model limitations.

4. The fine-tuned Multi-View Spatial Model (MVSM) demonstrates consistent and interpretable improvements across tasks and generalizes to external benchmarks (VSI-Bench, VSI-App). Additional analyses on backbone variation and answer formats support the robustness of the findings.

5. Overall, the paper is well-written and clearly presented, with informative figures, tables, and methodological descriptions that facilitate reproducibility.

**Weaknesses:**

1. The methodological novelty is limited. While the benchmark is comprehensive, its conceptual basis—evaluating and fine-tuning for 3D spatial reasoning—builds directly on existing work. The MVSM primarily extends a Qwen-VL baseline through additional spatially annotated data rather than introducing new model architectures or explicit 3D reasoning mechanisms.

2. The comparative analysis omits several relevant baselines, including specialized models such as SpatialVLM, SpatialPin, SpatialReasoner, Space-Qwen, and the Gemini 2.5 series, which would strengthen the empirical claims.

3. The paper’s assertion of being the “first comprehensive benchmark” is somewhat overstated, as prior works (e.g., 3DSRBench, SPHERE, All-Angles Bench) already address similar multi-view evaluation objectives. The contribution lies more in the dataset scale and integration than in conceptual originality.

**Questions:**

Please refer to the weakness

---

> ### Author Response · Authors · 2025-11-22
> **Response to Reviewer qX8B (Part 1)**
>
> # Weakness 1: Limited methodological novelty—mainly extends existing models with spatial annotations.
>
>  We appreciate the reviewer's assessment and clarify our specific contributions.
>
> **On Benchmark Contribution**
>
> ViewSpatial-Bench is **the most comprehensive benchmark for cross-viewpoint spatial localization evaluation**. Our motivation addresses a critical requirement for spatial intelligence: VLMs must understand not only their own spatial perception but also interpret spatial references from human collaborators' perspectives—essential for embodied AI and human-robot interaction.
>
> Existing benchmarks present scattered spatial questions from camera or human viewpoints separately, without comprehensive coverage of localization tasks. ViewSpatial-Bench provides **systematic joint evaluation across 5 localization task types and 18 directional categories**, enabling direct cross-perspective comparison and identification of critical failure patterns. **Our systematic framework enables discovery of counter-intuitive insights unobservable through scattered evaluations**: human-perspective reasoning (35.7%) unexpectedly outperforms camera-perspective tasks (33.2%), and models show opposite performance patterns across perspectives (Table 2: Object Orientation 26.9% camera vs 42.6% human)—indicating fundamental limitations in coherent cross-viewpoint spatial understanding.
>
> **On MVSM Training Approach**
>
> We acknowledge MVSM introduces no architectural innovations. Our methodological contribution is the **automated 3D annotation pipeline with perspective-aware dataset design**. The empirical finding is that targeted cross-viewpoint training suffices: consistent +41-46% improvements across three architectures (Appendix C.3) with generalization to downstream tasks, providing a scalable training paradigm for enhancing spatial localization capabilities without architectural modifications.
>
> # Weakness 2: The comparative analysis lacks several relevant specialized baselines, weakening the empirical claims.
>
> We appreciate the reviewer's suggestion and have conducted comprehensive evaluations on the suggested models, including both proprietary systems and specialized spatial reasoning models.
>
> **Updated proprietary model results (Table 2):**
>
> | Model          | Camera Tasks | Person Tasks | Overall |
> | -------------- | ------------ | ------------ | ------- |
> | GPT-5          | 43.48%       | 47.40%       | 45.49%  |
> | Gemini-2.5-pro | 43.52%       | 48.52%       | 46.09%  |
>
> **Specialized spatial reasoning models (Table 3):**
>
> | Model                 | Camera Tasks | Person Tasks | Overall |
> | --------------------- | ------------ | ------------ | ------- |
> | VST-7B-SFT            | 70.10%       | 58.07%       | 63.90%  |
> | VST-3B-SFT            | 46.05%       | 61.23%       | 53.87%  |
> | SpatialLadder-3B      | 39.62%       | 48.52%       | 44.21%  |
> | MindCube-3B-RawQA-SFT | 53.20%       | 28.07%       | 40.25%  |
> | SpaceQwen2.5-VL-3B    | 38.06%       | 41.46%       | 39.81%  |
>
> Models with dedicated spatial training demonstrate substantially better performance than general-purpose VLMs, with VST-7B-SFT achieving 63.90% overall accuracy. However, cross-viewpoint spatial localization remains challenging even for specialized spatial models. The inclusion of advanced proprietary models confirms our main findings hold consistently across diverse architectures. These results validate the necessity of comprehensive multi-perspective evaluation and confirm that perspective-taking in localization tasks represents a distinct challenge beyond general spatial reasoning.
>
> **We have included these comprehensive comparisons in the revised manuscript** with detailed analysis in the main results section.

---

> > ### Author Response · Authors · 2025-11-22
> > **Response to Reviewer qX8B (Part 2)**
> >
> > # Weakness 3: The "first comprehensive benchmark" claim is overstated given existing works; the contribution lies in dataset scale rather than conceptual originality.
> >
> > We appreciate the reviewer's clarification and agree that our contribution lies substantially in dataset scale and task integration. We acknowledge that prior benchmarks (3DSRBench, SPHERE, All-Angles Bench) have established important foundations in multi-view spatial evaluation.
> >
> > The specific distinction of ViewSpatial-Bench lies in its positioning as the first **comprehensive cross-viewpoint spatial localization** benchmark. Our motivation addresses spatial intelligence requirements: **VLMs must not only understand their own spatial perception but also** interpret spatial references from others' cognitive perspectives**—essential for embodied AI. **From the localization task perspective, **while existing benchmarks may include camera or human viewpoints, they present scattered basic spatial questions from each perspective without comprehensive localization task coverage.** As Table 1 demonstrates, 3DSRBench focuses on object-target queries without person-perspective evaluation, SPHERE and All-Angles Bench include limited directional categories (7 and 4 directions respectively). ViewSpatial-Bench provides systematic evaluation **across 5 localization task types and 18 directional categories**, enabling thorough cross-perspective assessment and detection of performance asymmetries (Table 2) within a unified framework.
> >
> > We will revise our manuscript to position ViewSpatial-Bench as providing **the most comprehensive joint evaluation of cross-perspective spatial localization** rather than claiming absolute primacy, while clearly acknowledging the foundational contributions of prior multi-view benchmarks.

---

> ### Author Response · Authors · 2025-11-27
> **Welcome Further Feedback**
>
> Dear Reviewer **qX8B**,
>
> We would like to thank you for the thoughtful and constructive feedback. During the rebuttal, we performed additional experiments and analyses and revised the manuscript accordingly to address your concerns. The experiments and analyses in our response are summarized as follows.
>
> - **Expanded baseline comparisons with specialized spatial reasoning models**
>   - We evaluated advanced proprietary models (GPT-5, Gemini-2.5-pro) and five specialized spatial models (VST-7B-SFT, VST-3B-SFT, SpatialLadder-3B, MindCube-3B-RawQA-SFT, SpaceQwen2.5-VL-3B).
>   - Results confirm our findings hold consistently across diverse architectures (Table 2 and Table 3).
> - **Clarified benchmark contribution and positioning**
>   - We repositioned ViewSpatial-Bench as providing the **most comprehensive joint evaluation of cross-perspective spatial localization** rather than claiming absolute primacy.
>   - We acknowledge foundational contributions of prior benchmarks while highlighting our systematic evaluation across 5 task types and 18 directional categories.
> - **Clarified methodological contribution of MVSM**
>   - We acknowledge MVSM introduces no architectural innovations—our contribution is the automated 3D annotation pipeline with perspective-aware dataset design.
>   - We demonstrate consistent +41-46% improvements across three architectures with downstream generalization (Appendix C.3).
>
> We are wondering if the responses have addressed your concerns, and would appreciate it if you considered raising your final rating. Looking forward to your further feedback!

---

### Author Response · Authors · 2025-11-27
**Summary of Responses to Reviewers**

We thank all reviewers for their valuable feedback and are encouraged that they found our problem formulation clear and well-motivated (qX8B, eAeM, FvxK, CiyB), recognizing the importance of cross-viewpoint spatial reasoning for embodied AI. Here, we provide a high-level summary of the changes we've made to address your concerns, and conclude with an overview of our key contributions.

Here is the summary of updates that we've made to address the reviewers' concerns:

- **We expanded baseline comparisons with specialized spatial reasoning models.** (qX8B)
  - We evaluated advanced proprietary models (GPT-5, Gemini-2.5-pro) and five specialized spatial models, confirming our findings hold consistently across diverse architectures (Table 2 and Table 3).
- **We provided comprehensive human annotation quality validation.** (eAeM)
  - We clarified that all 5,712 test samples underwent iterative human verification.
  - Human performance evaluation with 3 expert annotators achieved 92-94% accuracy with substantial inter-annotator agreement (κ_avg = 0.89).
- **We enhanced statistical rigor and experimental validation.** (eAeM, FvxK)
  - We added standard deviations across 5 independent runs, showing consistently low variance (±0.31 to ±0.71) and confirming result reliability (Table 7 in Appendix C.2).
  - Supplementary direct answer experiments validate MC format necessity, with models achieving only 0.31-15.06% exact match on compound directions.
- **We addressed generalization and overfitting concerns.** (CiyB, FvxK)
  - Task-specific OOD improvements reveal genuine transfer: Route Planning +9.54%, VSI-App +16.00% (Indoor: +23.00%; Outdoor: +9.00%).
  - Architecture-agnostic (+41-46% across three backbones) and format-agnostic performance confirm generalizable spatial learning.
- **We clarified benchmark positioning and contributions.** (qX8B, eAeM, CiyB)
  - We repositioned ViewSpatial-Bench as providing the most comprehensive joint evaluation of cross-viewpoint spatial localization.
  - We clarified that 3D coordinates are the core foundation of our annotation pipeline, and revised MVSM terminology to emphasize training approach rather than architectural innovation.

The contribution of our work is summarized as follows:

- **We introduce ViewSpatial-Bench, the most comprehensive benchmark for cross-viewpoint spatial localization**, systematically evaluating VLMs across 5 localization task types and 18 directional categories from both camera and human perspectives.
- **We develop an automated 3D spatial annotation pipeline** that efficiently generates large-scale, precisely annotated multi-view datasets with rigorous human verification.
- **We demonstrate that perspective-aware training effectively addresses cross-viewpoint spatial reasoning gaps**, achieving 46.24% improvement with strong generalization to downstream tasks.

Together, these contributions establish a crucial benchmark and training methodology for spatial intelligence in vision-language models, providing empirical evidence that modeling 3D spatial relationships from multiple perspectives enhances VLMs' spatial comprehension capabilities essential for embodied AI applications.

---

### Meta-Review · Area_Chair_5MpG · 2025-12-29

**Summary:**

This paper introduces ViewSpatial-Bench, a benchmark designed to evaluate how well VLMs understand spatial relationships from different perspectives, like distinguishing between a camera's view and a human's view. The authors construct a dataset using existing 3D scene collections and demonstrate that fine-tuning models on this data improves their performance. Despite the practical motivation to understand human perspectives, AC finds that the dataset largely repurposes existing datasets like ScanNet and MS-COCO without introducing a significant new data. As noted by the reviewers, the proposed model is essentially a standard backbone trained on specific data, which represents an engineering effort. Therefore, the paper is not ready for this conference, and AC believes that this work will benefit from more real-world data for the new benchmark.

**Reviewer Concerns:**

The authors provided answers during the discussion, such as explaining how they verified the data and adding the missing comparison models requested by Reviewer `qX8B`. However, problems remained regarding how the benchmark was built. Reviewer `eAeM` worried that because the answers were generated by computers using hidden data. Some questions might be impossible for a human or model to solve just by seeing the picture. Reviewer `FvxK` also questioned the long-term value of the project, noting that because a simple fine-tuning process improved the results so much, the task might be too easy or allow models to cheat rather than truly learning spatial reasoning. Reviewer `CiyB` added by noting that the model likely just memorized the training data.

**Reviewer Scores:**

Reviewer `eAeM` gave a score of 2 and is unlikely to change it, as the dataset questions were not realistic enough. Reviewer `qX8B` and Reviewer `FvxK` both gave borderline scores, acknowledging the work was sound but finding it lacked excitement because it felt more like an engineering project. Reviewer `CiyB` gave a score of 4, agreeing that the work was not very original compared to similar benchmarks. The split in scores reflects the general feeling that while the tool works, it does not offer enough new ideas to justify acceptance.

---

### Decision · Program_Chairs · 2026-01-26

Reject